# Targeting of SLC25A22 boosts the immunotherapeutic response in KRAS-mutant colorectal cancer

Qiming Zhou[1], Yao Peng[1,2], Fenfen Ji[1], Huarong Chen [1,3], Wei Kang [4], Lam-Shing Chan[1], Hongyan Gou[1], Yufeng Lin[1], Pingmei Huang[1], Danyu Chen[1], Qinyao Wei[1], Hao Su[1,3], Cong Liang[5], Xiang Zhang[1], Jun Yu [1] ✉ & Chi Chun Wong [1] ✉

KRAS is an important tumor intrinsic factor driving immune suppression in colorectal cancer (CRC). In this study, we demonstrate that SLC25A22 underlies mutant KRAS-induced immune suppression in CRC. In immunocompetent male mice and humanized male mice models, SLC25A22 knockout inhibits KRAS-mutant CRC tumor growth with reduced myeloid derived suppressor cells (MDSC) but increased CD8[+] T-cells, implying the reversion of mutant KRAS-driven immunosuppression. Mechanistically, we find that SLC25A22 plays a central role in promoting asparagine, which binds and activates SRC phosphorylation. Asparagine-mediated SRC promotes ERK/ETS2 signaling, which drives CXCL1 transcription. Secreted CXCL1 functions as a chemoattractant for MDSC via CXCR2, leading to an immunosuppressive microenvironment. Targeting SLC25A22 or asparagine impairs KRAS-induced MDSC infiltration in CRC. Finally, we demonstrate that the targeting of SLC25A22 in combination with anti-PD1 therapy synergizes to inhibit MDSC and activate CD8[+] T cells to suppress KRAS-mutant CRC growth in vivo. We thus identify a metabolic pathway that drives immunosuppression in KRAS-mutant CRC.

Colorectal cancer (CRC) is one of the most common cancers and the prognosis with metastatic CRC remains poor. Targeting of T-cell inhibitory molecules such as PD1 and CTLA-4 induce T-cell-mediated antitumor immunity with dramatic efficacies in hypermutated tumors with microsatellite instability-high (MSI-H) or DNA mismatch repair deficiency (dMMR)[1]. Nonetheless, this comprises a minor subset of CRC (~5%)[2], highlighting a need to improve response to ICB in CRC.

Activating KRAS mutations (~45%) are drivers of CRC. According to consensus molecular subtype (CMS) for CRC[3], KRAS mutation is associated with CMS3 subtype with poor immune infiltration. Mutant

KRAS is associated with reduced infiltration of lymphocytes in cancer genome atlas (TCGA) and KFSYSCC CRC datasets[4]. Mice with Kras[G12D] mutant CRC showed immunosuppression and resistance to anti-PD1 therapy[5]. This highlights the role of mutant KRAS in driving CRC progression by suppressing antitumor immunity.

KRAS is notoriously difficult to target with small molecules. Sotorasib, a first-in-class KRAS[G12C] inhibitor[6], failed to elicit benefit KRAS[G12C] CRC[7]. Our previous work has shown that SLC25A22[8,9], a member of mitochondrial transporter family (SLC25) that facilitates the transport of glutamate across the inner mitochondrial membrane into the mitochondrial matrix[10,11], functions as an oncogenic factor in

[1]Institute of Digestive Disease and Department of Medicine and Therapeutics, State Key Laboratory of Digestive Disease, Li Ka Shing Institute of Health Sciences, The Chinese University of Hong Kong, Hong Kong SAR, China. [2]Shenzhen University General Hospital, Shenzhen, China. [3]Department of Anaesthesia and Intensive Care and Peter Hung Pain Research Institute, The Chinese University of Hong Kong, Hong Kong SAR, China. [4]Department of Anatomical and Cellular Pathology, The Chinese University of Hong Kong, Hong Kong SAR, China. [5]State Key Laboratory of Cellular Stress Biology and School of Life Sciences, Xiamen University, Xiamen, China. ✉e-mail: junyu@cuhk.edu.hk; chichun.wong@cuhk.edu.hk

KRAS mutant CRC by generating metabolites critical for antioxidative defense[8] and epigenetic deregulation[9].

In this work, we show that SLC25A22 as an immunotherapeutic target in KRAS-mutant CRC. The knockout of SLC25A22 reverses KRAS-mediated immunosuppression by suppressing CXCL1, thereby impairing myeloid-derived suppressor cells (MDSC) recruitment and inducing cytotoxic T-cell activation. Finally, we demonstrate that siRNA nanoparticles targeting SLC25A22 synergizes with anti-PD1 to suppress tumor growth in KRAS-mutant CRC. Hence, we propose SLC25A22 as a therapeutic target for improving response to ICB therapy in KRAS-mutant CRC.

## Results

### Mutant KRAS represses antitumor immunity in CRC
To examine the role of mutant KRAS mutations in mediating immune landscape during colorectal tumorigenesis, we generated wildtype, KRAS mutant ($Kras^{G12D/+}Villin-Cre$), APC mutant ($Apc^{Min/+}$), and APC-KRAS mutant ($Apc^{Min/+}Kras^{G12D/+}Villin-Cre$) mice. We observed high mortality of APC-KRAS mutant mice with median survival of 45 days ($P < 0.0001$) (Fig. S1). At 45 days, wildtype or KRAS-mutant mice had no tumors; APC mutant and APC-KRAS mutant mice have $1.4 \pm 0.7$ and $18.0 \pm 8.4$ tumors in the colon, respectively (Figs. 1a, S1). RNA-seq of colon tumors of APC- and APC-KRAS mutant mice revealed that multiple pathway associated with immune response are enriched in APC-KRAS mutant mice including TNF and cytokine-cytokine receptor pathways (Fig. 1b). We thus hypothesized that mutant KRAS might modulate immune landscape in CRC. Consistently, flow cytometry revealed APC-KRAS mutant tumors had increased MDSC ($CD11b^+Gr1^+$), especially polymorphonuclear MDSC (PMN-MDSC) ($P < 0.01$) and the suppression of total T cells and $CD8^+$ T cells ($P < 0.05$) (Fig. 1c). $CD4^+$ T cells and macrophages were not altered. Our results indicate that KRAS mutation promotes immunosuppression.

### SLC25A22 loss abolishes mutant KRAS-induced immunosuppression in APC-KRAS mutant organoids and CT26 allograft models
We previously identified that SLC25A22 promotes KRAS-mutant CRC[8,9,12]. In light of the association of glutamine metabolism and antitumor immunity[13], we explored the role of SLC25A22 in KRAS-induced immunosuppression. CRC organoids from APC-KRAS mutant and $Apc^{min/+}Kras^{G12D/+}Slc25a22^{fl/fl}Villin-Cre$ mice with colon-specific Slc25a22 knockout (APC-KRAS-SLC25A22$^{KO}$) were implanted in immunocompetent mice (Fig. 1d). APC-KRAS-SLC25A22$^{KO}$ tumors exhibited arrested growth compared to APC-KRAS mutant tumors (Fig. 1d). Flow cytometry showed that total MDSC and PMN-MDSC were reduced by Slc25a22 knockout ($P < 0.05$); while total T-cell ($P < 0.05$), and $CD8^+$ T-cell ($P < 0.05$) were induced (Fig. 1e). Both immunofluorescence (IF) and immunohistochemistry (IHC) confirmed decrease of MDSC cells in Slc25a22 knockout tumors as exemplified by decreased $Cd11b^+Gr-1^+$ ($P < 0.05$) and $S100a8^+$ cells ($P < 0.01$), respectively (Fig. S2). IHC of CD8 ($P < 0.05$) validated increased T cells infiltration into SLC25A22 knockout tumors (Fig. S2). We validated these data in KRAS-mutant CT26 allografts (Fig. 1f). Slc25a22 knockout (Slc-KO) CT26 tumors showed lower tumor size (Fig. 1f) together with reduced MDSC ($P < 0.01$) and PMN-MDSC ($P < 0.01$), while total T cells and $CD8^+$ T cells were increased ($P < 0.01$) (Fig. 1g). These results imply that SLC25A22 knockout reversed KRAS-induced immunosuppression.

### SLC25A22 loss abrogates immunosuppression in spontaneous CRC
We retrospectively analyzed spontaneous colon tumors from APC-KRAS and APC-KRAS-SLC25A22$^{KO}$ transgenic mice[9]. MDSC was depleted in SLC25A22 knockout mice as shown by IF ($Cd11b^+Gr-1^+$, $P < 0.01$) and IHC (S100a8, $P < 0.05$), while $CD8^+$ T cells ($P < 0.05$) were induced (Fig. 1h), thus validating our observations in allografts.

### SLC25A22 loss abrogates mutant KRAS-induced immunosuppression in PBMC humanized mice
We next examined the effect of SLC25A22 in PBMC humanized mice model implanted with DLD1 cells with or without SLC25A22 knockout (Fig. 2a). SLC25A22 knockout (SLC-KO) impaired tumor growth ($P < 0.001$) and tumor weight ($P < 0.01$) (Fig. 2b). Analyses of tumor-infiltrating $huCD45^+$ immune cells confirmed reduction of human MDSC ($HLA-DR^-CD11b^+CD33^+$), together with increased infiltration of $CD8^+$ T cells (Figs. 2c and S3). This verified the effect of SLC25A22 on human $CD45^+$ immune cells infiltration in KRAS-mutant CRC.

### SLC25A22 is correlated with immunosuppression in human CRC
We next sought to validate the immunologic role of SLC25A22 in human CRC using a tissue microarray (TMA) cohort of 202 CRC patients. KRAS-mutant CRC accounts for half of CRC patients in this TMA cohort, and the majority of KRAS-mutant CRC also harbored mutations in APC (77/102, 75.4%). We performed co-staining of SLC25A22 and CD33, a marker for human MDSC[14,15], showing that cases with high SLC25A22 have increased infiltration of $CD33^+$ cells (Fig. 2d). SLC25A22 expression positively correlated with CD33 positive cells in KRAS-mutant CRC ($P < 0.05$), but not in KRAS-wildtype CRC (Fig. 2d). Accordingly, MDSC levels were elevated in SLC25A22-high compared to SLC25A22-low KRAS-mutant CRC ($P < 0.01$), but not in KRAS-wildtype CRC (Fig. 2d). Consistent results were found in TCGA cohort. We devised an MDSC score[16] based on COAD RNA-seq cohort (Fig. S4). MDSC-high cases have increased SLC25A22 mRNA compared to MDSC-low ones in KRAS-mutant CRC, but not in KRAS-wildtype CRC (Fig. 2e). As MDSC functions to antagonize $CD8^+$ T cells, we analyzed correlation between SLC25A22 and CD8 staining in our TMA cohort ($N = 202$) by IHC. By comparing KRAS wildtype and mutant CRC, we found that KRAS-mutant CRC have more cases with low CD8 (Fig. 2f), implying immunosuppression. Notably, SLC25A22-high CRC had lower CD8 scores compared to SLC25A22-low ($P < 0.001$) (Fig. 2g). Negative correlation between CD8 scores and SLC25A22 was identified in overall cohort ($\chi^2 = 13.8$; $P < 0.01$) or KRAS-mutant CRC ($\chi^2 = 6.6$; $P < 0.05$). Collectively, SLC25A22 is associated with an immunosuppressive phenotype in KRAS-mutant CRC.

### SLC25A22 knockout suppresses C-X-C chemokines in KRAS-mutant CRC
To probe the molecular mechanism of SLC25A22-mediated immunosuppression, we performed RNA-seq of SLC25A22 knockout DLD1 cells. Gene set enrichment analysis revealed 13 common dysregulated pathways in SLC25A22 knockout cells (Fig. 3a). Strikingly, top depleted pathway is Cytokine-Cytokine Receptor Interaction ($P < 0.0001$) (Fig. 3b, c), implying involvement of SLC25A22 in cytokine signaling in addition to its metabolic function.

We thus profiled mRNA expression of immune mediators by Inflammatory Response & Autoimmunity PCR array. Comparison of APC-KRAS mice tumors with adjacent normal tissues revealed that mRNA expression of cytokines-related genes was largely over-expressed in APC-KRAS tumors. In contrast, SLC25A22 knockout in DLD1 and CT26 cells showed an opposite trend with decreased cytokine expression (Fig. 3d). We identified CXCL1 and CXCL3 were up-regulated in APC-KRAS tumors, whilst being down-regulated by SLC25A22 knockout (Fig. 3d). qPCR validated down-regulated CXCL1/3 in KRAS-mutant CRC cells (DLD1, Colo26, and CT26) after SLC25A22 knockout (Fig. 3e). CXCL1/3 are chemokines that recruit MDSC. We further surveyed secreted cytokines in conditioned medium from DLD1-sgControl and DLD1-SLC-KO cells by Antibody Array (ab169812, Abcam) (Fig. 3f). Densitometry revealed CXCL1 and CXCL1/2/3 as the top depleted cytokines in SLC25A22 knockout cells (Fig. 3g). ELISA in DLD1, CT26 and Colo26 cells confirmed the down-regulated secretion of CXCL1/3 after SLC25A22 knockout (Fig. 3h).

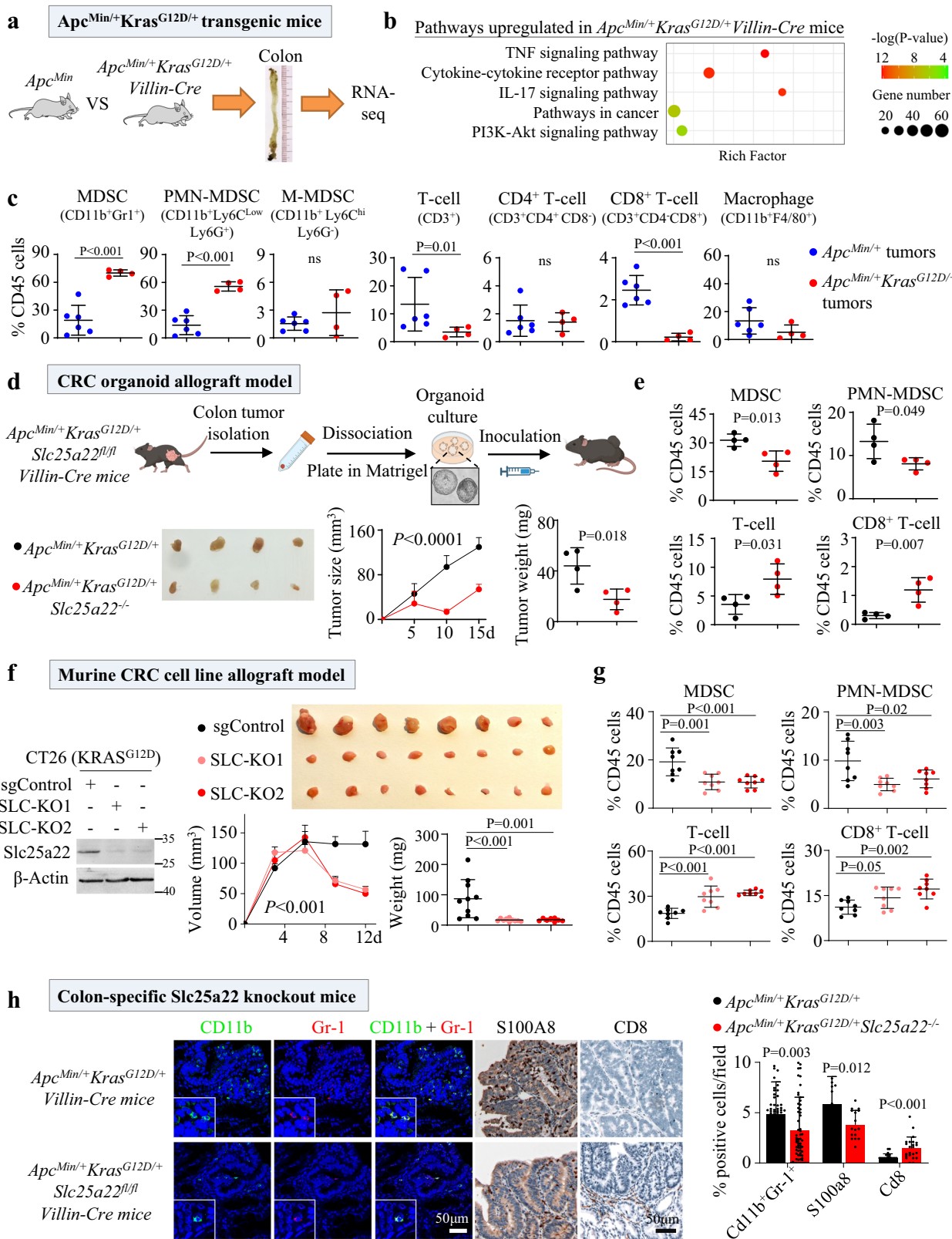

CXCL1 secretion was also suppressed in APC-KRAS-SLC25A22[KO] organoids compared to APC-KRAS organoids (Fig. S5) after normalization for cell viability. To validate if SLC25A22 modulates CXCL1/3 in vivo, we assessed their levels in serum and tumors from APC-KRAS organoid allografts and CT26 allografts models (Fig. 1). In CT26 allografts, SLC25A22 knockout reduced CXCL1 in serum ($P < 0.05$) and tumors ($P < 0.001$), whereas CXCL3 was down-regulated only in

tumors ($P < 0.01$) (Fig. 3i). APC-KRAS-SLC25A22[KO] organoids engrafted mice showed decreased serum CXCL1, while CXCL1/3 were reduced in tumors ($P < 0.001$) compared to APC-KRAS organoids (Fig. 3i). Corroborating our findings, CXCL1/3 are top cytokines correlated with SLC25A22 mRNA expression ($P < 0.0001$) in TCGA cohort (Fig. 3j). Collectively, SLC25A22 mediates CXCL1/3 in KRAS-mutant CRC.

**Fig. 1 | SLC25A22 knockout affects immune cell infiltration in KRAS-mutant CRC. a** Breeding of $Apc^{Min/+}$ and $lsl$-$Kras^{G12D/+}Villin$-$Cre$ to give $Apc^{Min/+}Kras^{G12D/+}$ $Villin$-$Cre$ mice. At 7 weeks, colon tumors were harvested for RNA-seq. **b** Immune-related signaling are enriched in colon tumors ($n = 2$) from $Apc^{Min/+}Kras^{G12D/+}Villin$-$Cre$ compared to $Apc^{Min/+}$ mice ($n = 2$). Rich factor is defined as the ratio of differentially expressed gene number to total gene number in a particular pathway. *P* value, Fisher's exact test. **c** Flow cytometry of immune cell populations in $Apc^{Min/+}Kras^{G12D/+}Villin$-$Cre$ ($n = 4$) and $Apc^{Min/+}$ mice ($n = 6$). MDSC, including PMN-MDSC, were increased in $Apc^{Min/+}Kras^{G12D/+}Villin$-$Cre$ mice, together with decreased total T cells and CD8$^+$ T cells in tumors. Each dot represents an independent mouse. **d** Colon tumor organoids derived from $Apc^{Min/+}Kras^{G12D/+}$ or $Apc^{Min/+}Kras^{G12D/+}Slc25a22^{-/-}$ mice with intestinal Slc25a22 knockout were implanted into C57BL/6 mice. Tumor growth was inhibited in Slc25a22 knockout organoids ($n = 4$). Each dot represents an independent mouse.

**e** Slc25a22 knockout inhibited MDSC and PMN-MDSC, but induced total and CD8$^+$ T cells ($n = 4$). Each dot represents an independent mouse. **f** SLC25A22 knockout (Slc-KO) in KRAS-mutant CT26 cells inhibited tumor growth in BALB/c mice (sgControl and SLC-KO1: $n = 10$ per group; SLC-KO2: $n = 8$). Each dot represents an independent tumor. **g** SLC25A22 knockout inhibited MDSC and PMN-MDSC, but increased total T cells and CD8$^+$ T cells ($n = 8$). **h** Immunofluorescence and immunohistochemistry of immune cells in the tumors from $Apc^{Min/+}Kras^{G12D/+}$ and $Apc^{Min/+}Kras^{G12D/+}Slc25a22^{-/-}$ mice ($n = 6$), confirming reduced MDSC but increased CD8$^+$ T cells by SLC25A22 knockout. Each dot represented the value from an independent capture field. Data are shown as mean ± SD (**c**–**h**) and ± SEM for the growth curves (**d**, **f**). Two-tailed Student's *t* test (**c**–**e**, **h**). Two-tailed one-way ANOVA (**f**, **g**). Two-tailed two-way ANOVA for growth curve (**d**, **f**). ns, no significance. Source data are provided as a Source Data file.

## SLC25A22 recruits MDSC via CXCL1-CXCR2 axis and promotes activation of MDSC in KRAS-mutant CRC

As SLC25A22 knockout reduced MDSC infiltration, we hypothesized that SLC25A22 is required for tumor-induced MDSC chemotaxis through CXCL1/3. We respectively isolated splenic MDSC from murine allograft implanted mice and human MDSC from PBMC humanized mice (Fig. 4a) for transwell migration assays. We next examined effect of CRC-conditioned medium with or without SLC25A22 knockout. Conditioned medium from control cells promoted MDSC migration ($P < 0.01$), an effect blunted in SLC25A22 knockout cells ($P < 0.05$) (Fig. 4b), indicating that SLC25A22 loss impaired the ability of CRC cells to promote MDSC chemotaxis. To ask if CXCL1 or CXCL3 is key for MDSC chemotaxis, we knockdown CXCL1/3 in DLD1 cells (Fig. S6). CXCL1-siRNA, but not CXCL3-siRNA, suppressed migration of MDSC in DLD1-sgControl conditioned medium ($P < 0.01$) (Fig. 4c). siCxcl1 also suppressed MDSC migration in conditioned medium from CT26-sgControl and Colo26-sgControl cells ($P < 0.05$), while it had no effect on MDSC migration in conditioned medium from SLC25A22 knockout cells (Fig. 4c). Anti-CXCL1 antibody also abrogated MDSC migration in conditioned medium from CT26-sgControl ($P < 0.05$) and Colo26-sgControl ($P < 0.05$) cells, but not in Slc-KO cells (Fig. 4d). Reciprocally, exogenous CXCL1 restored MDSC migration in CT26-Slc-KO ($P < 0.05$) and Colo26-Slc-KO ($P < 0.05$) cell conditioned medium (Fig. 4e). SLC25A22-mediated CXCL1 is thus critical for recruitment of MDSC. CXCR2 is the receptor for CXCL1 expressed on MDSC[17]. In line with this notion, two CXCR2 inhibitors SB265610 and SX682 abolished MDSC migration in DLD1-sgControl and CT26-sgControl conditioned medium (Fig. 4f) but had no effect on SLC-KO cells (Fig. S7). We next analyzed the correlation between Cxcl1 and MDSC in our in vivo models. We observed positive correlations between CXCL1 and MDSC in tumors from APC-KRAS organoid allografts ($R = 0.686$, $P < 0.0001$), CT26 allografts ($R = 0.602$, $P < 0.01$), and DLD1 xenografts in humanized mice ($R = 0.454$, $P < 0.05$) (Fig. 4g). SLC25A22 thus promotes MDSC chemotaxis via a CXCL1-CXCR2 axis in KRAS-mutant CRC.

MDSC inhibits T-cell activation by secreting arginase-1 (ARG1) and indolamine-2,3-dioxygenase (IDO) that deplete amino acids[18] as well as immunosuppressive molecules (TGFβ, PD-L1)[19]. We, therefore, analyzed MDSC markers in CT26 allografts. Compared to CT26-sgControl, PD-L1 ($P < 0.001$), iNOS ($P < 0.001$), and ARG1 ($P < 0.05$) mRNA were down-regulated in MDSC from SLC25A22 knockout CT26 allografts (Fig. 4h). Flow cytometry validated that surface PD-L1 protein in MDSC from CT26 SLC25A22 knockout tumors was reduced compared to control tumors (Fig. 4i).

We next investigated the functional role of SLC25A22-mediated recruitment of MDSC in KRAS-mutant CRC. We evaluated if blockade of CXCR2 modulated SLC25A22-mediated KRAS-mutant CRC growth by treatment of CT26 allografts with SB265610 (3 mg/kg). SB265610 inhibited growth of CT26-sgControl allografts, as evidenced by reduced tumor volume ($P < 0.001$) and weight ($P < 0.05$), but had no effect on Slc-KO tumors (Fig. 4j). Similarly, SB265610 suppressed APC-KRAS organoid allografts growth, while having no effect on APC-KRAS-

SLC25A22$^{KO}$ counterparts (Fig. S8). In both models, SB265610 suppressed MDSC infiltration in controls but not in SLC25A22 knockout tumors, and intratumoral MDSC correlated with tumor burden ($P < 0.05$) (Figs. 4k and S8). This supports the role of SLC25A22 in MDSC recruitment and activation in KRAS-mutant CRC to facilitate tumor growth.

## SLC25A22-driven MDSC suppresses T-cell proliferation and function to facilitate tumorigenesis in KRAS-mutant CRC

MDSC-mediated immunosuppression is involved in inactivation of cytotoxic CD8$^+$ T cells. As SLC25A22 knockout inhibited MDSC infiltration and function, we performed T-cell suppression assays (Figs. 5a and S9). MDSC derived from SLC25A22 knockout CT26 tumors have attenuated ability to inhibit T-cell proliferation compared to control tumors (Fig. 5b). MDSC from SLC25A22 knockout tumors were deficient in inhibiting CD8$^+$ T-cell functional markers IFN-γ, TNF-α and Granzyme B as compared to control tumors (Fig. 5c). In vivo, MDSC levels negatively correlated with CD8$^+$ T cells in APC-KRAS organoid allografts ($R = -0.814$; $P < 0.0001$), CT26 allografts ($R = -0.770$; $P < 0.0001$) and DLD1 xenografts ($R = -0.486$; $P = 0.01$) in PBMC humanized mice (Fig. 5d), consistent with blockade of T-cell proliferation by SLC25A22-mediated MDSC. In APC-KRAS organoid allografts (Fig. 5e) and CT26 allografts (Fig. 5f), knockout of SLC25A22 elevated IFN-γ ($P < 0.001$), TNF-α ($P < 0.01$), and Granzyme B ($P < 0.001$) expression in CD8$^+$ T-cell compared to their respective controls. Accordingly, CD8$^+$ T-cell depletion by anti-CD8 abrogated growth inhibitory effect of SLC25A22 knockout in CT26 allografts (Fig. 5g), verifying the function of CD8$^+$ T-cell-driven antitumor immunity in growth inhibition by SLC25A22 knockout. Taken together, SLC25A22 knockout impaired MDSC infiltration and function, which reinvigorated CD8$^+$ T-cell growth and cytotoxicity, leading to tumor suppression.

## SLC25A22-mediated glutamine metabolism drives CXCL1 chemokine via asparagine biosynthesis

We next sought to decipher the mechanism through which SLC25A22 mediates CXCL1 expression. SLC25A22 is a mediator of glutamine metabolism[8,9]. Here, we focused on CXCL1, as it is the key cytokine involved in MDSC recruitment. To assess if CXCL1 depends on glutamine, we incubated DLD1-sgControl and DLD1-SLC-KO cells in escalating glutamine (0–4 mM) and determined CXCL1 expression. CXCL1 mRNA was increased by glutamine in a dose-dependent manner (Fig. 6a). ELISA validated that glutamine increased CXCL1 secretion in DLD1 and CT26 cells (Fig. 6b). SLC25A22 loss abrogated CXCL1 mRNA and secretion at high glutamine ($P < 0.001$), while having a lesser effect at low glutamine (Fig. 6a, b), implying that glutamine-induced CXCL1 depends on SLC25A22.

Mitochondrial glutamate is catabolized through glutamate dehydrogenase (GDH) or transaminase to enter TCA cycle[20]. We treated DLD1 cells with inhibitors of GDH (R162/purpurin) or transaminase (aminooxyacetate, AOA) and evaluated their effects on CXCL1. AOA

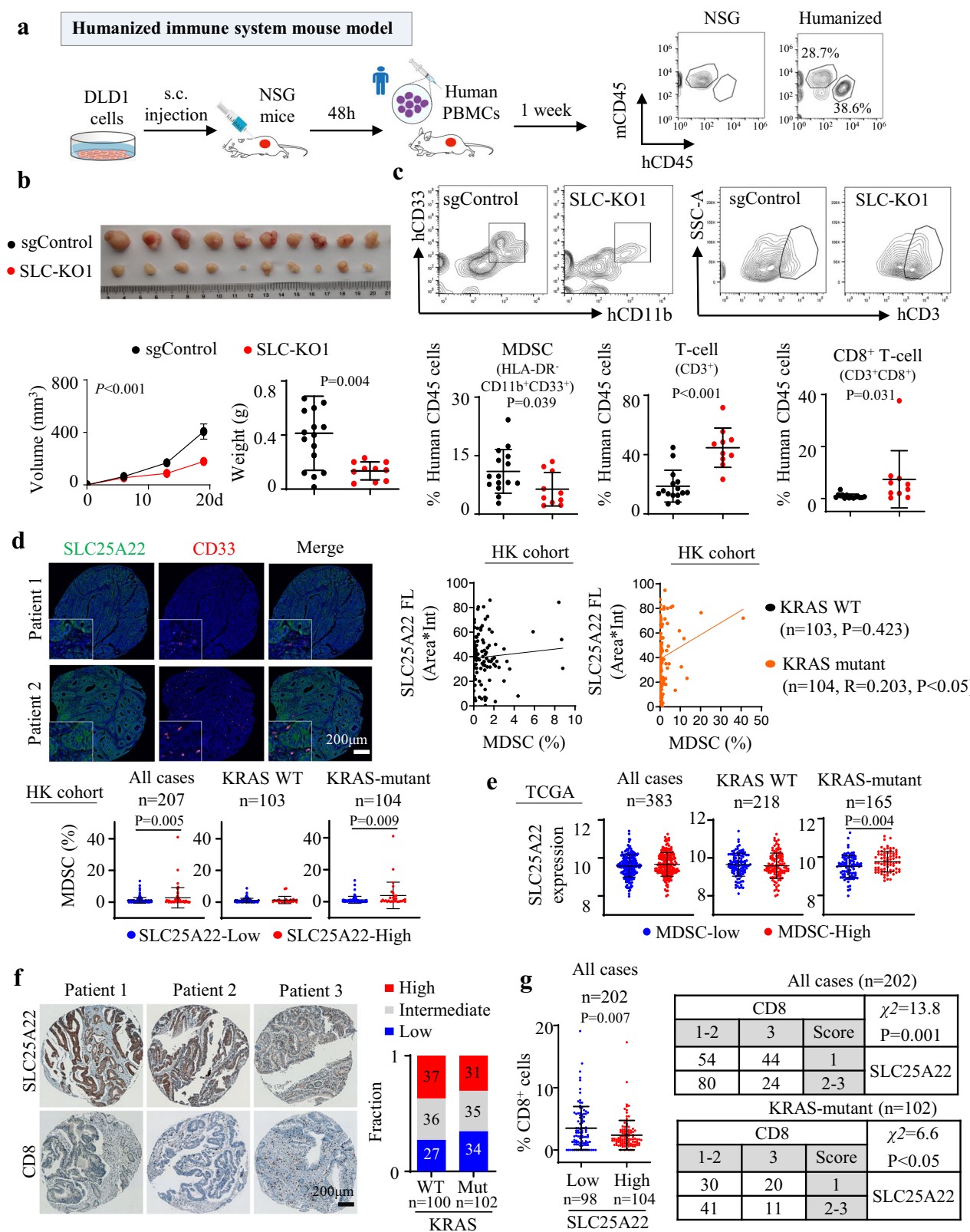

reduced CXCL1 mRNA and secretion ($P < 0.001$) and GDH inhibitors exerted an opposite effect ($P < 0.001$), implying that glutamine may activate CXCL1 via transaminase (Fig. 6c). Transaminase generates non-essential amino acids in addition to TCA cycle metabolites (Fig. 6d). We hence incubated DLD1-sgControl and DLD1-SLC-KO cells with downstream metabolites. Only asparagine rescued CXCL1 mRNA levels in both DLD1 and CT26 cells with SLC25A22 knockout

($P < 0.001$), while having no such effects on respective sgControl cells (Figs. 6e and S10a, b). Consistently, asparagine restored the secretion of CXCL1 in both DLD1 ($P < 0.05$) and CT26 ($P < 0.01$) cells with SLC25A22 knockout (Fig. 6f), but not in sgControl counterparts (Fig. S10c). Other metabolites failed to rescue CXCL1 and had no effect on intracellular asparagine (Fig. S10d). Ectopic expression of SLC1A3 to increase aspartate uptake, on the other hand, enabled the recuse of

**Fig. 2 | SLC25A22 correlates with immune suppression in PBMC humanized mice and human KRAS-mutant CRC patients. a** The construction and validation of PBMC humanized mice model. **b** SLC25A22 knockout significantly inhibited the growth of DLD1 xenografts in PBMC humanized mice (sgControl: $n = 15$; SLC-KO1: $n = 10$). Each dot represents an independent tumor. **c** SLC25A22 knockout inhibited tumor infiltration of human MDSC (HLA-DR$^-$CD11b$^+$CD33$^+$) while increasing T-cell (hCD3$^+$CD4$^+$/ hCD3$^+$CD8$^+$) infiltration (sgControl: $n = 15$; SLC-KO1: $n = 10$). Each dot represents an independent tumor. **d** Co-immunofluorescence of SLC25A22 and CD33 in a tissue microarray CRC cohort ($n = 207$). Positive correlation was revealed in only KRAS-mutant CRC (*upper*). Increased MDSC infiltration was found in SLC25A22-high CRC with mutant KRAS (*lower*). Each dot represents an independent patient. **e** In TCGA cohort ($n = 383$), SLC25A22 mRNA expression is significantly higher in MDSC-high tumors compared to MDSC-low tumors in KRAS-mutant CRC. Each dot represents an independent patient. **f** Immunohistochemistry of CD8 and SLC25A22 in CRC tissue microarray ($n = 202$). CD8$^+$ T cells were reduced in KRAS-mutant CRC compared to wildtype CRC patients. **g** High SLC25A22 expression correlated with reduced CD8$^+$ T-cell in KRAS-mutant CRC (All cases: $n = 202$; KRAS-mutant: $n = 102$). Each dot represents an independent patient. Data are shown as mean ± SD (**b, c, d, e, g**) and ± SEM for growth curve **b**. Two-tailed Student's *t* test (**b, c, d, e, g**). Two-tailed two-way ANOVA for growth curves (**b**). Chi-Square test (**g**) and Pearson correlation test (**d**). Source data are provided as a Source Data file.

CXCL1 mRNA and secretion by extracellular aspartate (Fig. 6e–f). We next determined dose-dependent effect of asparagine, revealing that low-dose asparagine (25 μM) could restore intracellular asparagine (Fig. S11a) and CXCL1 mRNA and secretion (Fig. S11b) of CT26-Slc-KO cells to that of CT26-sgControl cells. Collectively, these results implied asparagine might be a metabolite linking SLC25A22 and CXCL1.

To verify that SLC25A22 knockout caused asparagine depletion, we performed [$^{13}C_5$]-glutamine labeling (Fig. 6g). Indeed, both total and labeled asparagine was reduced by SLC25A22 knockout in DLD1 and CT26 cells ($P < 0.001$). SLC25A22 knockout had no effect on ASNS catalytic activity (Fig. S11c), suggesting that it reduced asparagine biosynthesis by lowering intracellular aspartate levels[8]. Further, exogenous asparagine restored ability of DLD1-SLC-KO and CT26-Slc-KO cells to induce MDSC migration (Fig. 6h). If asparagine is important for SLC25A22-mediated CXCL1 secretion and MDSC recruitment, asparagine depletion would phenocopy the effects of SLC25A22 depletion. We thus depleted asparagine by asparagine synthetase (ASNS) knockdown (Fig. 6i) or Asparaginase (ASNase) treatment (Fig. 6j). siASNS or ASNase suppressed CXCL1 mRNA ($P < 0.01$) and secretion ($P < 0.01$) in DLD1 and CT26 cells (Fig. 6k–l). Consequently, conditioned medium from siASNS or ASNase-treated cells showed impaired capacity to induce MDSC migration (Fig. 6m). This support the hypothesis that SLC25A22-induced asparagine underlies increased CXCL1 secretion and MDSC recruitment in KRAS-mutant CRC.

### SRC kinase is as an asparagine sensor kinase that drives ETS2-mediated CXCL1 transcription in KRAS-mutant CRC

We next delineated the signaling linking asparagine to CXCL1. Phosphokinase array after short-term asparagine stimulation revealed SRC phosphorylation (Fig. 7a). In addition, SLC25A22 knockout inhibited p-SRC (Fig. 7b), and p-SRC also responded to glutamine in a SLC25A22-dependent manner that was rescued by asparagine (Fig. 7c), validating asparagine as a key metabolite in SLC25A22-dependent SRC activation. Asparagine has been shown to bind LCK[21], a member of SRC kinase family. We thus asked if asparagine could directly interact with SRC. Thermal shift assay revealed that asparagine increased thermal stability of recombinant SRC (Fig. 7d). BIAcore assay with recombinant SRC showed that asparagine interacts with SRC with binding affinity ($K_d$) of 21 μM (Fig. 7e). Moreover, asparagine induced the phosphorylation and kinase activity of SRC recombinant protein (Fig. 7f). D-Asparagine, on the other hand, failed to bind to or activate SRC (Fig. S12). Against this backdrop, we hypothesize that SRC kinase function as a sensor of SLC25A22-asparagine axis.

CXCL1 is primarily regulated by transcription. To reveal transcription factors that link SLC25A22 to CXCL1, we performed in silico analysis of CXCL1 promoter (−2000 to 100 bp) and overlapped the candidates with that downstream of SRC and SLC25A22, identifying ETS2 as a potential hit (Fig. 7g). Western blot validated that SLC25A22 knockout in DLD1 and CT26 cells suppressed ETS2 and p-ETS2 (active) expression (Fig. 7h), and ETS2 nuclear translocation (Fig. 7i). It is noteworthy that ETS2 protein was induced by glutamine in an SLC25A22-dependent fashion (Fig. 7j), mirroring that of CXCL1. Asparagine rescued ETS2 in SLC25A22-depleted cells (Fig. 7j). In

contrast, siASNS inhibited ETS2 (Fig. 7k), confirming that ETS2 responds to asparagine levels. To verify that ETS2 promotes CXCL1 transcription, we performed ETS2 knockdown in DLD1 and CT26 cells (Fig. S13). siETS2 reduced CXCL1 mRNA and secretion (Fig. 7l), while ETS2 overexpression restored CXCL1 mRNA and secretion in SLC25A22 knockout DLD1 and CT26 cells (Fig. 7m). In silico prediction[22–24] indicates that ETS2 binds to CXCL1 promoter (Fig. 7n). Chromatin immunoprecipitation-PCR validated binding of ETS2 to CXCL1 promoter (Fig. 7n). Furthermore, ETS2 activated CXCL1 promoter activity, as determined by luciferase assay (Fig. 7o). Finally, the activation of ETS2 by SLC25A22 depends on SRC, as SRC inhibitor abrogated p-ETS2 and ETS2, and CXCL1 secretion in DLD1 cells (Fig. 7p). These data indicate that SRC functions as an asparagine sensor that drives ETS2/CXCL1 axis downstream of SLC25A22.

### SLC25A22 knockout synergizes with anti-PD1 therapy to induce T-cell activation and suppress KRAS-mutant CRC growth

As SLC25A22 knockout repressed MDSC, we hypothesize that targeting SLC25A22 may synergize with ICB therapy. We first explored the therapeutic effect of anti-PD1 in combination with SLC25A22 knockout in CT26 cells and MC38 cells overexpressing mutant Kras[5]. In CT26 and MC38-Kras$^{G12V}$ allografts, anti-PD1 alone had no effect on tumor growth (Fig. 8a, b), in line with the immune resistance phenotype of KRAS-mutant CRC[5]. Single SLC25A22 knockout suppressed tumor growth by ~50% in CT26 and MC38-Kras$^{G12V}$ allografts ($P < 0.05$). Importantly, SLC25A22 knockout plus anti-PD1 exerted synergistic effects as compared to single treatment ($P < 0.01$), leading to tumor regression and reduction in tumor weight (Fig. 8a, b). Flow cytometry revealed SLC25A22 knockout plus anti-PD1 synergized to reduce MDSC and promote cytotoxic CD8$^+$ T-cell infiltration and IFN-γ expression (Fig. 8c, d). Moreover, we established orthotopic allograft model with TP53-null APC-KRAS and APC-KRAS-SLC25A22$^{KO}$ organoids (Fig. S14a). Consistently, SLC25A22 knockout synergized with anti-PD1 to inhibit tumor growth (Fig. S14b). MDSC was reduced by SLC25A22 knockout, while its combination with anti-PD1 synergistically induced CD8$^+$ T-cell infiltration and IFN-γ expression (Fig. S14c). These findings indicated SLC25A22 is a promising target that sensitizes KRAS-mutant CRC to ICB therapy.

### Targeting SLC25A22 with nanoparticles small interfering RNA synergized with anti-PD1 to suppress the growth of KRAS-mutant CRC

siRNA nanoparticles are an attractive form of biocompatible therapy targeting specific genes for depletion. Here, we utilized nanoparticle (VNP) for siRNA delivery in vivo[25]. SLC25A22 siRNA was modified by 2'-O-methyl groups to improve stability, and then incorporated into nanoparticles. We tested therapeutic efficacy of SLC25A22 siRNA in combination with anti-PD1 in MC38-Kras$^{G12V}$ allografts. When tumor volume reached ~100 mm$^3$, mice were randomized to receive VNP-siControl, VNP-siSLC25A22 with or without anti-PD1. Combination of VNP-siSLC25A22 and anti-PD1 synergistically inhibited growth of MC38-Kras$^{G12V}$ allografts ($P < 0.05$) (Fig. 8e). VNP-siSLC25A22 also potentiated effect of anti-PD1 in promoting cytotoxic IFN-γ$^+$ CD8$^+$

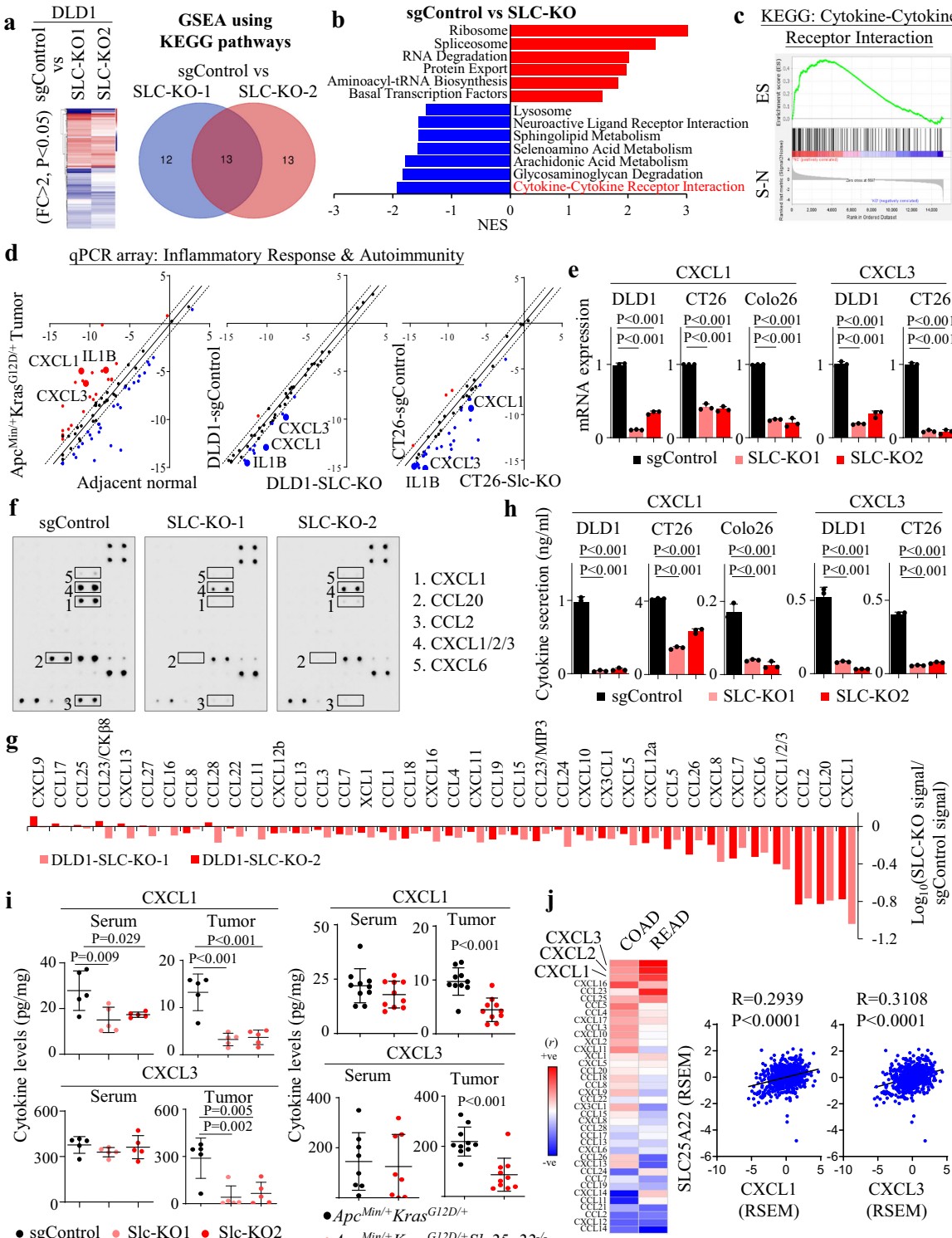

**Fig. 3 | SLC25A22 knockout abrogates secretion of CXCL1 and CXCL3 in vitro and in vivo. a** RNA-seq of SLC25A22 knockout DLD1 cells and gene set enrichment analysis (GSEA) for the identification of common differentially regulated pathways in SLC-KO1 and SLC-KO2 cells ($n = 4$). **b, c** GSEA enrichment scores for differentially regulated gene sets unveiled the cytokine-cytokine receptor interaction signaling pathway as the top pathway depleted in SLC25A22 knockout cells. **d** Inflammatory Response and Autoimmunity PCR array showed that CXCL1, CXCL3 and IL1B were induced in $Apc^{Min/+}Kras^{G12D/+}Villin-Cre$ mice tumors, but were down-regulated by SLC25A22 knockout (FC > 2). **e** qPCR validated that SLC25A22 knockout inhibited CXCL1/3 mRNA in DLD1, CT26 and Colo26 cells ($n = 3$). Each dot represents an independent sample. **f** Antibody array showed SLC25A22 knockout down-regulated

cytokine secretion in DLD1 cells. **g** Densitometry showed CXCL1 and CXCL1/2/3 were top-down-regulated cytokines. **h** ELISA confirmed SLC25A22 knockout impaired CXCL1/3 secretion in DLD1 (72 h), CT26 (24 h) and Colo26 (24 h) ($n = 3$). Each dot represents an independent sample. **i** Detection of CXCL1/3 in serum and tumors of mice implanted with CT26 allografts (*left*, $n = 5$) and $Apc^{Min/+}Kras^{G12D/+}$ organoid allografts (*right*, $n = 10$) with or without SLC25A22. Each dot represents an independent mouse. **j** SLC25A22 mRNA correlates with CXCL1/2/3 mRNA in TCGA CRC (COAD-READ) cohort ($n = 677$). Each dot represents an independent patient. Data are shown as mean ± SD (**e, h, i**). Two-tailed one-way ANOVA (**e, h, i**). Two-tailed Student's $t$ test analysis for two-group comparison **i**. Pearson correlation test **j**. Source data are provided as a Source Data file.

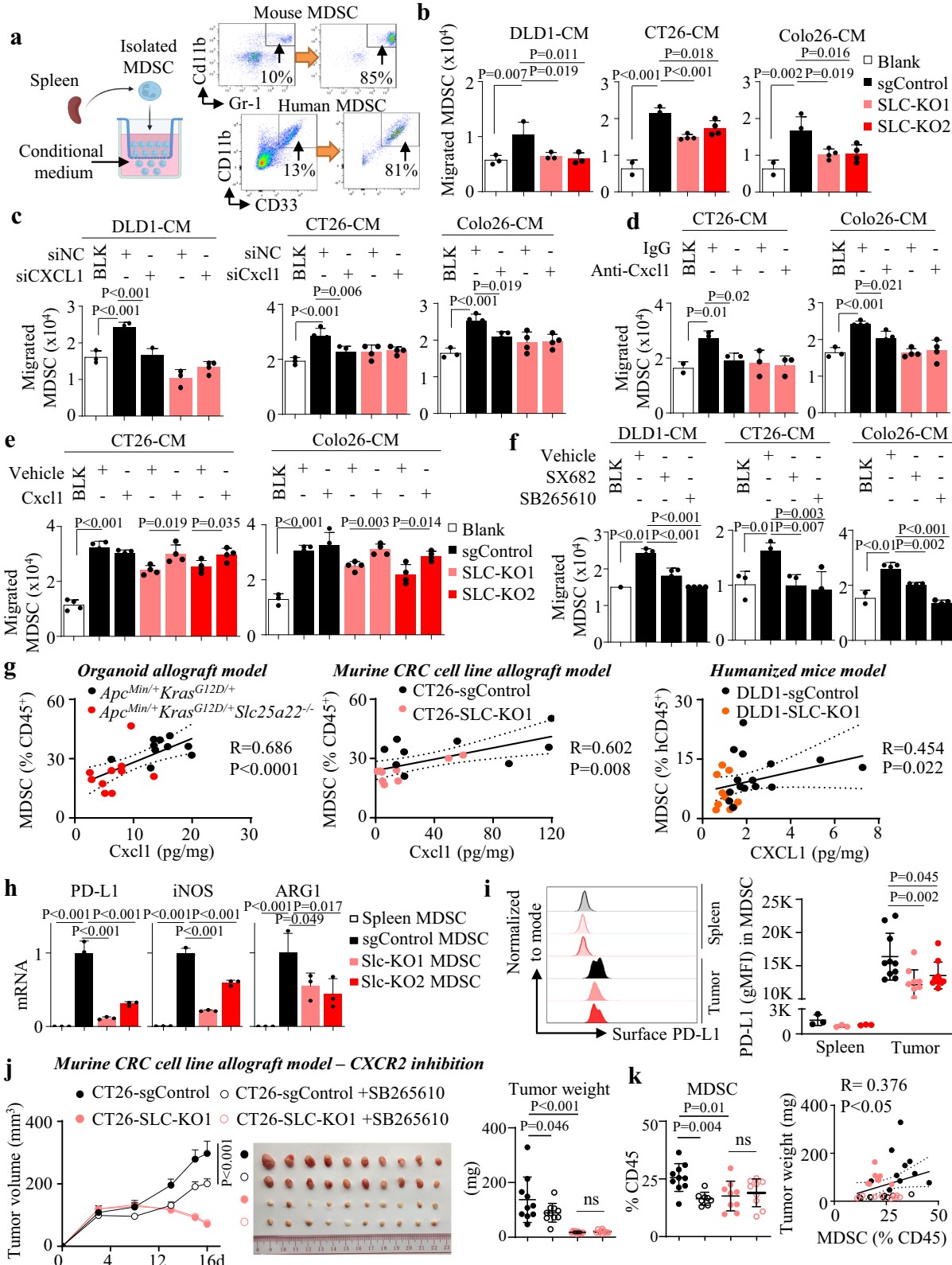

T-cell (Fig. 8f). siSLC25A22 plus anti-PD1 is a promising therapeutic approach for CRC. Finally, we demonstrated that targeting downstream signaling of SLC25A22 via SRC inhibitor (Fig. S15) or ASNase (Fig. S15) potentiated the effectiveness of anti-PD1 in inhibiting the growth of MC38-Kras^G12V allografts, in agreement with the SLC25A22-asparaingine-SRC axis as a therapeutic target for overcoming immunotherapy resistance in KRAS-mutant CRC.

## Discussion

Oncogenic KRAS mutations are common in CRC. Mutant KRAS is associated with an immunosuppressive microenvironment in CRC[26]. We sought to decipher the molecular mechanism of mutant KRAS-mediated immunosuppression. A key hallmark of KRAS-mutant CRC is metabolic reprogramming to meet bioenergetic, biosynthetic, and redox demands of tumor cells. Here, we reveal that SLC25A22, a

**Fig. 4 | SLC25A22 knockout abrogates recruitment and activation of MDSC in KRAS-mutant CRC via a CXCL1-CXCR2 axis. a** Flow cytometry validation of isolated mouse MDSC (CD11b⁺ Gr-1⁺) and human MDSC (CD11b⁺CD33⁺) from the spleens of tumor-implanted syngeneic mice and PBMC humanized mice, respectively. **b** Conditioned medium from sgControl cells induced MDSC migration, which was impaired by SLC25A22 knockout (*n* = 4). Each dot represents an independent sample. **c** siCXCL1, but not siCXCL3, abrogated induction of MDSC migration in sgControl cell conditioned medium (*n* = 4). Each dot represents an independent sample. **d** Anti-Cxcl1 neutralizing antibody (0.25 μg/mL) suppressed MDSC migration in sgControl cell conditioned medium; but had no effect on SLC25A22 knockout cells (CT26: *n* = 3; Colo26: *n* = 4). Each dot represents an independent sample. **e** Recombinant Cxcl1 (1 ng/ml) rescued MDSC migration in CT26-Slc-KO and Colo26-Slc-KO conditioned medium (*n* = 4). Each dot represents an independent sample. **f** CXCR2 inhibitors SX682 (2 μM) or SB265610 (10 μM) abolished MDSC migration in sgControl cell conditioned medium (*n* = 4). Each dot represents an independent sample. **g** Tumoral CXCL1 positively correlated with MDSC infiltration in *Apc^Min/+Kras^G12D/+* organoids

(*n* = 10) in C57BL/6 mice, CT26 allografts (*n* = 9) in BALB/c mice, and DLD1 xenografts (sgControl, *n* = 15; SLC-KO1, *n* = 10) in PBMC humanized mice. Each dot represents an independent mouse. **h** qPCR of isolated MDSC from CT26 allografts showed that activation markers PD-L1, iNOS, and ARG1 were enhanced in intratumoral MDSC compared to that of splenic MDSC, and was abrogated in SLC25A22 knockout tumors (*n* = 3). Each dot represents an independent mouse. **i** Flow cytometry of MDSC from CT26 allografts mice showed that surface PD-L1 protein on MDSC were induced in tumors compared with spleen, which was impaired in SLC25A22 knockout tumors (spleen: *n* = 3; tumors: *n* = 10). Each dot represents an independent sample. **j** SB265610 suppressed the growth of CT26-sgControl allografts (*n* = 10) and **k** downregulated MDSC (*n* = 9), but had no effect on CT26-Slc-KO allografts in BALB/c mice. MDSC positively correlated with tumor weight (*n* = 36). Each dot represents an independent tumor. Data are shown as mean ± SD. Two-tailed one-way ANOVA (**b** −**d**, **f**, **h**−**k**). Two-tailed Student's *t* test analysis for two-group comparison **e**. Spearman correlation test (**g**, **k**). Source data are provided as a Source Data file.

mitochondrial glutamate carrier, is involved in maintaining an immune repressive microenvironment in KRAS-mutant CRC, and that targeting SLC25A22 could re-activate antitumor immunity and synergize with anti-PD1 to induce tumor regression.

Comprehensive evaluation of tumor-infiltrating immune cells revealed that SLC25A22 knockout consistently decreased infiltration of immunosuppressive MDSC in KRAS-mutant tumors in murine allografts, intestine-specific SL25A22 knockout mice, and in human CRC xenografts engrafted in PBMC humanized mice, whereas CD4⁺ and CD8⁺ T cells were increased. In human KRAS-mutant CRC patients, SLC25A22 expression was positively and negatively correlated with MDSC and CD8⁺ T-cell tumor infiltration, respectively. These data implied that SLC25A22, a mitochondrial glutamate transporter, is involved in the crosstalk between glutamine metabolism and compromised antitumor immunity in KRAS-mutant CRC.

Chemokines comprise a large member of small chemotactic cytokines that orchestrate recruitment and trafficking of immune cells via interaction with their cognate receptors. RNA-seq revealed that cytokine-cytokine receptor pathway is depleted by SLC25A22 knockout in KRAS-mutant CRC cells. Integrated analysis demonstrated that knockout of SLC25A22 down-regulated multiple chemokines, especially CXCL1. Importantly, CXCL1 is known to recruit MDSC by binding to their CXCR2 receptor[27]. As a consequence, SLC25A22 knockout impaired the capacity of KRAS-mutant CRC cells to induce MDSC chemotaxis. Corroborating in vitro results, CXCL1 were consistently reduced in serum and tumors of mice inoculated with SLC25A22 knockout cells, and was correlated with reduced infiltration of MDSC. Besides, SLC25A22 knockout impaired MDSC activity and PD-L1 expression. Accordingly, MDSC from SLC25A22 null tumors were deficient in T-cell suppression, culminating in increased CD8⁺ T-cell infiltration and re-activation. Blockade of MDSC with CXCR2 inhibitors repressed MDSC infiltration and reversed SLC25A22-dependent CRC allograft growth in vivo. Our results underscore that CXCL1-CXCR2 axis mediated MDSC infiltration is critically involved in SLC25A22-driven tumorigenesis in KRAS-mutant CRC.

Having shown that SLC25A22 is integral to immunosuppression in KRAS-mutant CRC via MDSC recruitment, we next questioned if SLC25A22-immunity crosstalk is a consequence of metabolic rewiring. SLC25A22, a mitochondrial glutamate transporter, is involved in glutaminolysis[8,9]. Intriguingly, we demonstrated that CXCL1 was dose-dependently activated by glutamine in KRAS-mutant CRC cells. Also noteworthy is that SLC25A22 knockout impaired CXCL1 under glutamine replete conditions but not in its absence, implying that SLC25A22 as the link between glutamine rewiring and chemokine secretion in KRAS-mutant CRC. Oh, et al.[28] also showed that the glutamine antagonist JHU083 represses CSF3-driven MDSC infiltration, supporting our finding that glutamine promotes chemokine signaling to attract MDSC, possibly in a tumor-specific manner. Beyond MDSC, others have

reported alternative roles of glutamine in regulating antitumor immunity in cancer. JHU083, for instance, enhances oxidative phosphorylation and antitumor activity of CD8⁺ T cells[13], and it also synergizes with immunotherapy to achieve total tumor ablation[29]. V-9302, an inhibitor of glutamine uptake and utilization, suppressed PD-L1 expression in tumor cells, thus promoting anti-PD-L1-mediated antitumor activities of CD8⁺ T cells[30]. In contrast, glutaminase inhibitor CB-839 exerts off-target effects by impairing glutamine metabolism in CD8⁺ T cells and leading to T-cell suppression[31]. The role of SLC25A22-driven glutamine metabolism in the CXCL1-MDSC crosstalk affirms function of tumoral glutamine metabolism in promoting immunosuppression.

We identified downstream metabolites and molecular mechanism responsible for the crosstalk between SLC25A22 and CXCL1. Asparagine supplementation rescued secretion of CXCL1 in DLD1 and CT26 cells with SLC25A22 knockout. Further, the direct manipulation of cellular asparagine by either ASNS knockdown or Asparaginase treatment both impaired CXCL1 expression and secretion, leading to reduced capacity to promote MDSC migration in vitro, suggesting that asparagine is the key metabolite mediating SLC25A22-induced CXCL1 secretion and subsequently the recruitment of MDSC. We demonstrate that SRC, a tyrosine-protein kinase, functions as an asparagine sensor downstream of SLC25A22. Asparagine directly binds and activates SRC phosphorylation, subsequently promoting ETS2. ETS2 mediates CXCL1 transcription. Collectively, we identified a SLC25A22-asparagine-SRC/ETS2 signaling in driving CXCL1 expression and immunosuppression in KRAS-mutant CRC.

Asparagine is a non-essential amino acid, and it functions as a double-edged sword in tumorigenesis. On one hand, asparagine is an essential regulator of cancer cell amino acid homeostasis, anabolic metabolism, proliferation and tumor metastasis[32,33]. On the contrary, availability of extracellular asparagine was involved in optimal T-cell effector responses[34]. An key difference, however, is that KRAS-mutant CRC cells biosynthesize asparagine de novo via SLC25A22[8], whereas lymphocytes satisfy its demand largely through extracellular uptake[35]. Targeting of SLC25A22 therefore selectively abrogates asparagine-CXCL1-MDSC axis in tumors whereas sparing antitumor immune cells, thereby creating a therapeutic window for intervention. Hence, targeting of glutamine metabolism is a promising strategy for simultaneouly suppressing tumorigenesis and eliciting a potent antitumor immune response. However, as extracellular asparagine can rescue CXCL1 expression in SLC25A22-depleted tumors, this approach might require further modulation of plasma asparagine levels in vivo.

KRAS-mutant CRC is associated with immunologically "cold" tumors and resistance to ICB therapy in CRC patients.[36] Mutant KRAS-mediated MDSC recruitment is an obstacle to ICB therapy effectiveness by suppressing an effective antitumor cytotoxic T-cell response. Here, we showed that SLC25A22 knockout synergized with anti-PD1 therapy to mediate tumor regression in KRAS-mutant CRC allograft

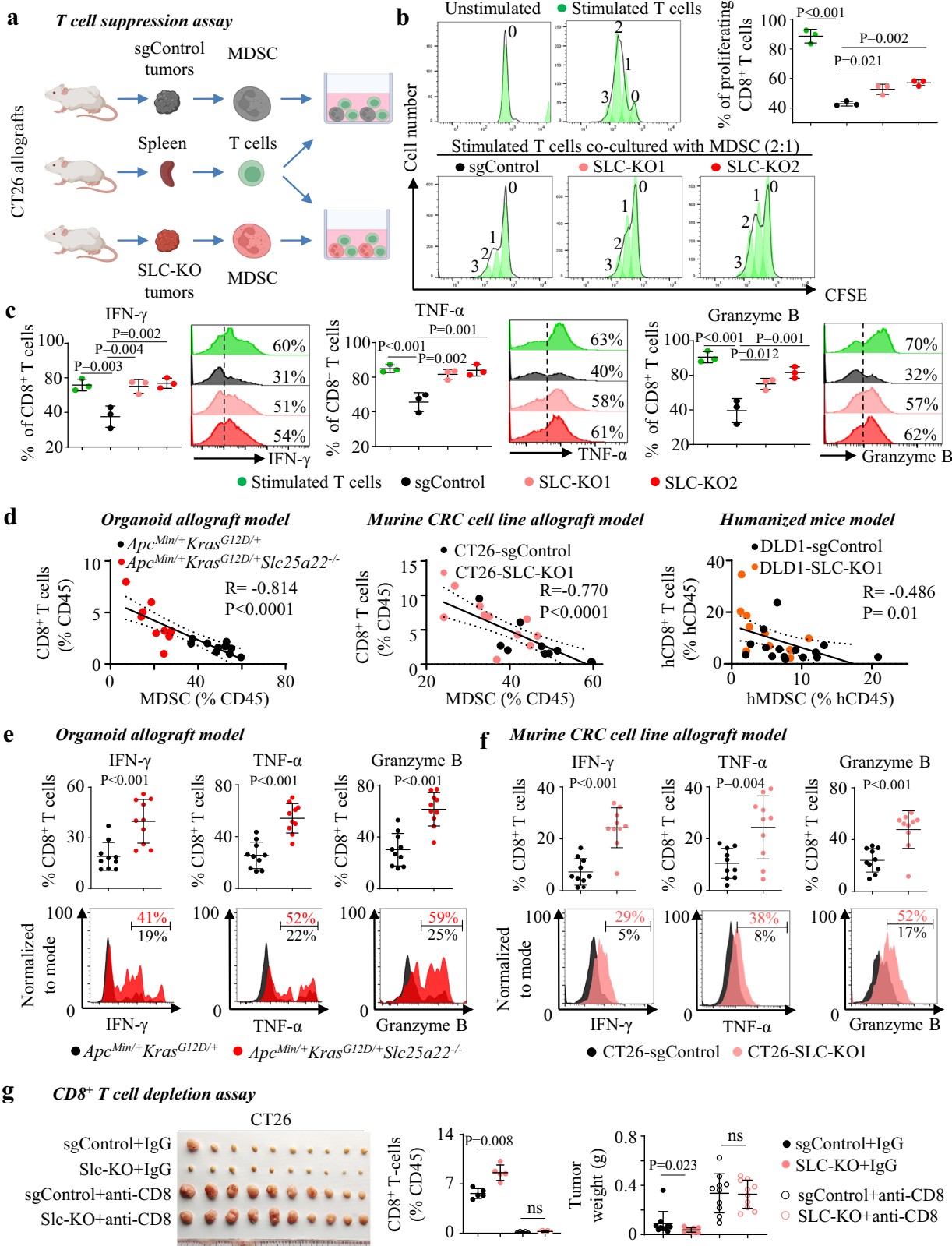

**Fig. 5 | SLC25A22-mediated MDSC recruitment represses T-cell proliferation and activation. a** Workflow of T-cell suppression assay. **b** T-cell suppression assay showed that MDSC isolated from CT26-Slc-KO allografts have attenuated ability to suppress T-cell proliferation (*n* = 3) and **c** expression of cytotoxic markers IFN-γ, TNF-α, and Granzyme B (*n* = 3). Each dot represents an independent sample. **d** Tumoral MDSC negatively correlated with CD8⁺ T cells in *Apc^{Min/+}Kras^{G12D/+}* allografts (*n* = 10), CT26 allografts (*n* = 10), and DLD1 xenografts (sgControl, *n* = 15; SLC-KO, *n* = 10). Each dot represents an independent tumor. **e** In *Apc^{Min/+}Kras^{G12D/+}* organoid allografts (*n* = 10) and **f** CT26 allografts (*n* = 10), SLC25A22 knockout increased

the tumor-infiltrating CD8⁺ T cells expressing activation marker (IFN-γ, TNF-α, and Granzyme B). Each dot represents an independent tumor. **g** CD8⁺ T cells depletion with anti-CD8 antibody abrogated growth inhibition by SLC25A22 knockout in CT26 allograft (*n* = 10). Each dot represents an independent mouse (*left*) or tumor (*right*). Data are shown as mean ± SD (**b**, **c**, **e**–**g**). Two-tailed one-way ANOVA (**b**, **c**). Two-tailed Student's *t* test for two-group comparison (**e**, **f**). Two-tailed Mann–Whitney *U* test **g**. Two-tailed Pearson correlation test **d**. ns, no significance. Source data are provided as a Source Data file.

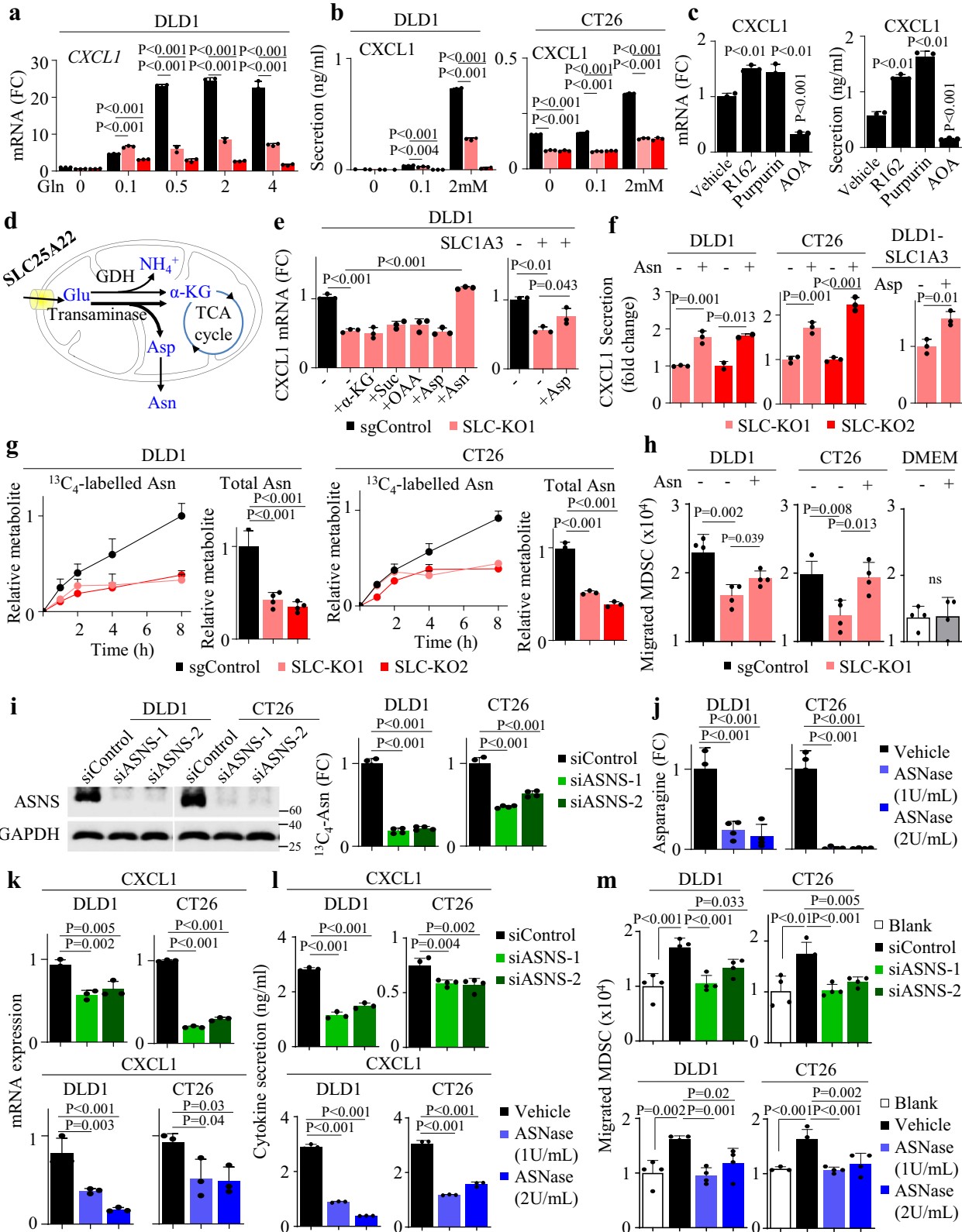

models, accompanied by synergistic induction of cytotoxic CD8[+] T-cell activation. In addition, we designed a SLC25A22-targeting VNP system for in vivo siRNA delivery[37], and showed that it potentiates anti-PD1 efficacy in KRAS-mutant CRC by rejuvenating cytotoxic CD8[+] T cells in tumor microenvironment. Together, VNP-siSLC25A22 is a potential therapeutic option for sensitizing KRAS-mutant CRC to ICB therapy.

In summary, we defined a metabolic circuitry of SLC25A22-asparagine-SRC/ ETS2/CXCL1 promoting an immunosuppressive microenvironment in KRAS-mutant CRC. SLC25A22-mediated CXCL1 recruits MDSC and inactivates cytotoxic T cells (Fig. 8G). Our work implies strategies targeting SLC25A22 to improve therapeutic response to immunotherapies in KRAS-mutant CRC.

**Fig. 6 | SLC25A22 drives glutamine-dependent CXCL1 via asparagine. a** In glutamine-depleted (12 h) DLD1 cells, addition of glutamine (72 h) dose-dependently induced CXCL1 mRNA. SLC25A22 knockout selectively inhibited CXCL1 mRNA at high glutamine levels ($n = 3$). Each dot represents an independent sample. **b** After glutamine depletion (12 h), glutamine (72 h) dose-dependently induced CXCL1 secretion in DLD1 cells and was impaired by SLC25A22 knockout ($n = 3$) (*left*). In glutamine-depleted (12 h) CT26 cells, glutamine (24 h) dose-dependently induced Cxcl1 secretion. SLC25A22 knockout inhibited Cxcl1 secretion at high glutamine ($n = 3$) (*right*). Each dot represents an independent sample. **c** Transaminase inhibition by aminooxyacetate (AOA, 100 μM) inhibited CXCL1 mRNA; whereas blockade of glutamate dehydrogenase 1 (GDH1) by Purpurin (25 μM) and R162 (25 μM) had an opposite effect ($n = 3$). A similar effect was observed for CXCL1 secretion ($n = 3$). Each dot represents an independent sample. **d** Schematic diagram showing the metabolic outputs of glutamine via SLC25A22. **e** Supplementation of downstream metabolites (2 mM, 24 h) (α-KG, dimethyl α-ketoglutarate; Succ, dimethyl succinate; OAA, dimethyl oxaloacetate; Asp, aspartate; Asn, asparagine) showed that asparagine restored CXCL1 mRNA in SLC25A22 knockout DLD1 cells ($n = 3$). Aspartate restored CXCL1 mRNA expression in DLD1-SLC-KO cells overexpressing SLC1A3 ($n = 3$). Each dot represents an independent sample. **f** Asparagine (2 mM, 24 h) restored CXCL1 secretion in both DLD1 and CT26 cells with knockout of SLC25A22 ($n = 3$).

Aspartate restored CXCL1 secretion in DLD1-SLC-KO cells overexpressing SLC1A3 ($n = 3$). Each dot represents an independent sample. **g** After depleting glutamine (12 h), [$^{13}C_5$]-Glutamine stable isotope labeling and LC-MS was performed in DLD1 and CT26 cells with or without SLC25A22 knockout. Total and $^{13}$C-labeled asparagine were reduced by SLC25A22 knockout ($n = 3$). Each dot represents an independent sample. **h** Treatment with asparagine (2 mM, 24 h) restored the capacity of DLD1-SLC-KO and CT26-Slc-KO cell conditioned medium to promote MDSC migration, without direct effects on MDSC migration ($n = 4$). **i** Validation of ASNS knockdown by western blot (*left*). LC-MS of DLD1 and CT26 cells co-transfected with siASNS and SLC1A3, followed by incubation with $^{13}C_4$-aspartate (2 mM, 96 h) confirmed the ASNS blockade ($n = 4$). Each dot represents an independent sample. **j** LC-MS showed that Asparaginase (ASNase) (24 h) depleted cellular asparagine in DLD1 and CT16 cells ($n = 4$). Each dot represents an independent sample. **k** siASNS (*upper*) or ASNase (*lower*) reduced CXCL1 mRNA and **l** secretion in DLD1 and CT26 cells ($n = 3$). Each dot represents an independent sample. **m** Conditioned medium from siASNS (*upper*) or ASNase-treated (*lower*) DLD1 and CT26 cells had reduced ability to induce MDSC migration. ($n = 4$). Each dot represents an independent sample. Data are shown as mean ± SD (**a**–**c**, **e**–**m**) and ± SEM for metabolite curves **g**. Two-tailed one-way ANOVA (**a**–**c**, **e**, **g**–**m**). Two-tailed Student's $t$ test **f**. ns, no significance. Source data are provided as a Source Data file.

# Methods

## Study approvals

For human samples and donors, informed consent was obtained for all patients. This study was approved by Clinical Research Ethics Committee of the Chinese University of Hong Kong. All the animal work is approved by the Animal Experimentation Ethics Committee of the Chinese University of Hong Kong.

## Clinical samples

Tissue microarray of 202 human CRC samples were collected from the patients in the Prince of Wales Hospital, Hong Kong. Informed consent for the study specimens were obtained from all subjects. This study was approved by the Clinical Research Ethics Committee of the Chinese University of Hong Kong.

## Cell culture

293T, CRC cells line DLD1 (MSI-H), CT26 (MSS) and MC38 (MSI-H) were obtained from American Type Culture Collection (Rockville, MD). Colo26 cells (MSS) were provided by Dr. Yujuan Dong from the Chinese University of Hong Kong. All cells were cultured in Dulbecco's modified Eagle medium (DMEM) with 10% Fetal Bovine Serum and 1% penicillin-streptomycin at 37 °C in a 5% $CO_2$ humidified atmosphere.

## MDSC isolation and in vitro migration assay

Murine MDSC were isolated from spleens of allograft implanted mice using EasySep™ Mouse MDSC (CD11b⁺Gr1⁺) Isolation Kit (#19867, STEMCELL Technologies). Human MDSC (CD11b⁺CD33⁺) were sorted by FACS from the spleens of PBMC humanized mice bearing DLD1 xenografts. To perform migration assay, freshly isolated MDSC ($5 \times 10^4$/well) were seeded into upper chambers 24-well transwell plates (8.0 μm; Costar Corning). Filtered conditioned medium (CM) from DLD1, CT26, or Colo26 cells with or without SLC25A22 knockout were added to lower chambers as chemoattractant. After 4 h incubation, migrated cells in the bottom chamber were counted. In some assays, recombinant CXCL1/3 (1 ng/ml, lower chamber, R&D Systems), anti-CXCL1 antibody (0.25ug/ml, lower chamber, R&D Systems), SX682 (2 μM, upper chamber, Selleck) or SB265610 (10 μM, upper chamber, Tocris Bioscience) were added.

## T-cell suppression assay

MDSC were isolated from allograft tumors inoculated in BALB/c mice. T cells were separated from splenocytes of a wildtype BALB/c mice mouse using a mouse Pan-T-cell isolation kit (#19851, STEMCELL

Technologies) and labeled with CFSE (Thermo Fisher). MDSC and T cells were co-cultured in 96-well plates with Dynabeads Mouse T-Activator CD3/CD28 (#11456D, Thermo Fisher) at an MDSC/T-cell ratio of 1:2. CFSE fluorescence was quantified after 72 h by flow cytometry, with peaks identified by proliferation modeling tool in FlowJo software.

## Immunohistochemistry and Immunofluorescence

Paraffin-embedded tissues were sectioned, deparaffinized with xylene and rehydrated in alcohol. Antigen retrieval was performed in microwave with Tris/EDTA buffer (pH 9.0) for 30 min. Hydrogen peroxide (3%) was used to block the endogenous peroxidase activity. Slides were incubated with primary antibodies overnight, including anti-CD4 (1:2000, ab183685, Abcam), anti-CD8 (1:2000, ab217344, Abcam), anti-S100A8 (1:800, 15792-1-AP, Proteintech), or anti-SLC25A22 (HPA014662; Sigma-Aldrich). After washing, slides were incubated with rabbit on rodent HRP-polymer (RMR622L, BioCare Medical) for 40 min and then developed with DAB Plus (ThermoFisher). For IF, paraffin-embedded sections were co-incubated with anti-CD11b (1:500, ab133357, Abcam) and anti-Gr-1 (1:100, #108448, BioLegend) overnight, followed by secondary antibodies Goat anti-Rabbit IgG (H + L) Cross-Adsorbed Secondary Antibody, Alexa Fluor™ 488 (Invitrogen, #A-11008, Polyclonal, 1:1000) and Goat anti-Rat IgG (H + L) Cross-Adsorbed Secondary Antibody, Alexa Fluor™ 594 (Invitrogen, #A-11007, Polyclonal, 1:1000). Slides were stained with DAPI/mounting medium (ThermoFisher) and examined by fluorescence microscopy (Leica Las X v5.02, Leica).

## ELISA determination of CXCL1 and CXCL3

CXCL1 and CXCL3 were determined in cell supernatant, serum, and tumor tissues. Cell supernatant was collected and filtered by 0.45 mm strainer. Tumour tissues was weighed, grinded, and treated with Cell Extraction Buffer PTR (Abcam) containing a protease inhibitor cocktail (Roche, Switzerland). The serum was separated from blood of tumor bearing mice. Mouse CXCL1 (ab216951, Abcam), mouse CXCL3 (ab272191, Abcam), human CXCL1 (ab190805, Abcam) or human CXCL3 (ab234574, Abcam) ELISA kits were performed according to the manufacturer's instructions.

## Flow cytometry

For flow cytometry, single cells from tissues were stained with indicated antibodies (Table S2) for 30 min at 4 °C (BioLegend). For intracellular marker staining, cells were incubated for 4 h in the presence of PMA (30 ng/ml), Ionomycin (1ug/ml), and monensin (2.5ug/ml). Following the

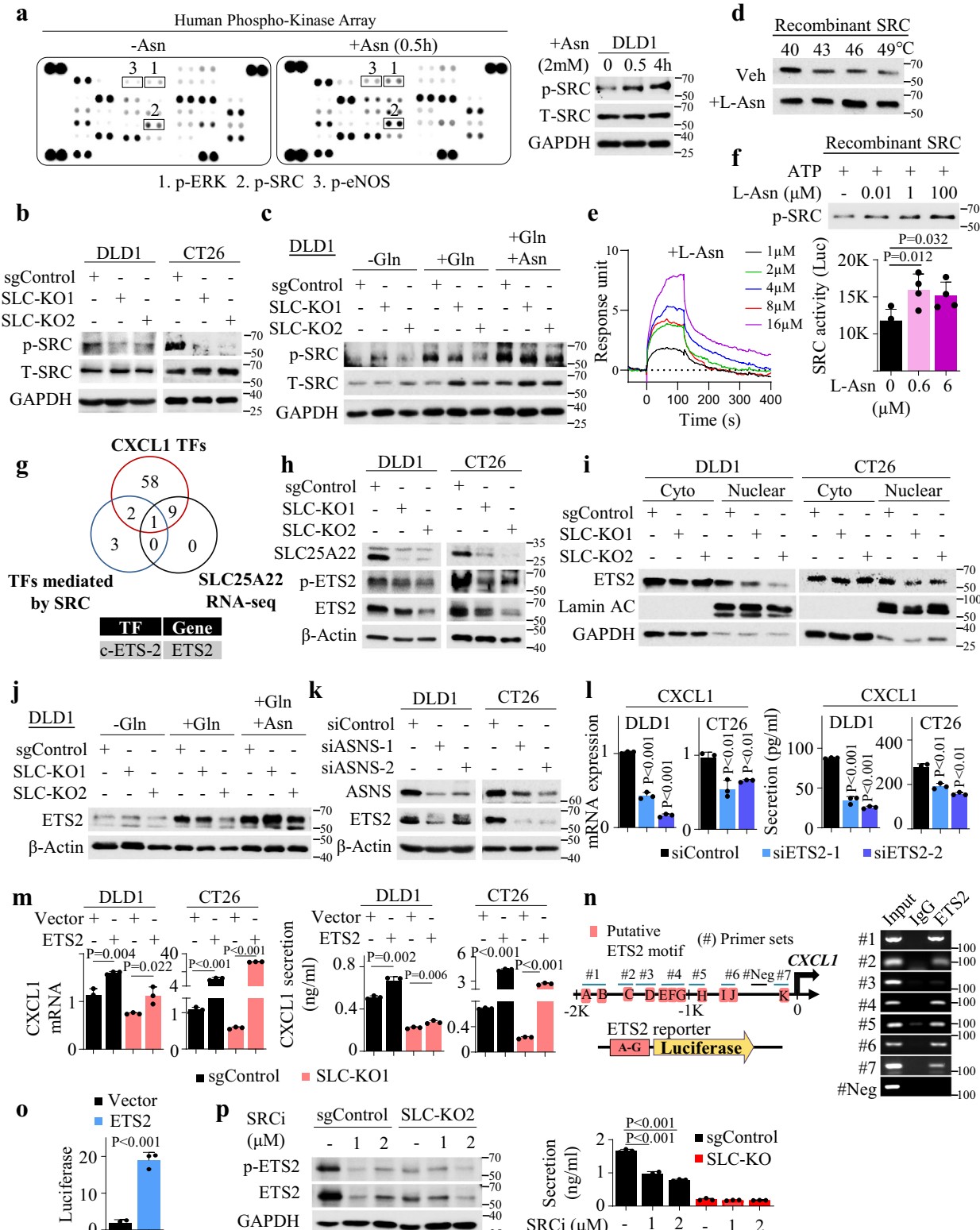

extracellular marker staining and fixation, cells were permeabilized (#00-8333-56, eBioscience) and stained with antibodies against intracellular markers. Flow cytometry was performed on BD FACSCelesta (BD Biosciences), and data were analyzed by FlowJo software (FlowJo v10).

## Tumor models and treatments
Male C57BL/6 J mice (6–8 weeks old) were implanted subcutaneously with APC-KRAS or APC-KRAS-SLC25A22$^{KO}$ organoids ($3 \times 10^5$ cells/tumor). Male BALB/c mice (6–8 weeks old) were subcutaneously

injected with CT26-Slc-KO or CT26-sgControl cells ($2 \times 10^6$ cells/tumor). Tumor size was measured by a digital calliper. Tumor volume was calculated as follow: volume = length × width$^2$ × 0.5. SB265610 (R&D systems) treatment was initiated one day after tumor cell inoculation (3 mg/kg body weight, i.p., 6 times/week). Anti-CD8 (150 µg/mice, i.p., BE0061, Bio X Cell) were injected 1 day before tumour cells inoculation, and twice a week thereafter. Anti-PD1 (200 µg per mice, i.p., BE0146, Bio X Cell) treatment was initiated after randomization at a frequency of three times per week. All mice

**Fig. 7 | SLC25A22-asparagine axis is sensed by SRC to activate the ETS2/ CXCL1 signaling in KRAS-mutant CRC. a** Phospho-Kinase Array (*n* = 1 experiment) revealed that asparagine addition (2 mM, 0.5 h, 4 h) in glutamine-low (0.1 M, 12 h) medium activated p-SRC and validation by Western blot (*n* = 3 independent biological replicates). **b** SLC25A22 knockout suppressed p-SRC in DLD1 and CT26 cells (*n* = 2). **c** Glutamine (1 mM, 12 h) promoted p-SRC in an SLC25A22-dependent manner. Asparagine (2 mM, 12 h) restored p-SRC in SLC25A22 knockout cells (*n* = 3 independent biological replicates). **d** Asparagine increased thermal stability (5 min incubation) of recombinant SRC (*n* = 3 independent biological replicates). **e** BIAcore analysis of the binding of asparagine to recombinant SRC (*n* = 2). **f** Asparagine-induced recombinant SRC phosphorylation and kinase activity (*n* = 4). Each dot represents an independent sample. **g** Overlapping of in silico prediction of CXCL1 transcription factors (TFs), SRC-downstream TFs, and SLC25A22 knockout RNA-seq data revealed ETS2 as a putative transcription factor for CXCL1. **h** SLC25A22 knockout suppressed ETS2 and p-ETS2 expression, and **i** ETS2 nuclear localization in DLD1 and CT26 cells (*n* = 2). **j** Glutamine (1 mM, 12 h) increased ETS2 protein in a SLC25A22-dependent manner. Asparagine (2 mM, 12 h) rescued ETS2 in SLC25A22 knockout cells (*n* = 2). **k** ASNS knockdown suppressed ETS2 expression (*n* = 2). **l** ETS2 knockdown reduced CXCL1 mRNA and secretion (*n* = 3). Each dot represents an independent sample. **m** ETS2 overexpression rescued CXCL1 mRNA and secretion in SLC25A22 knockout cells (*n* = 3). Each dot represents an independent sample. **n** In silico identification of ETS2 binding sites on CXCL1 promoter and validation by ChIP-PCR. **o** Luciferase reporter assay confirmed that ETS2 activated transcription of CXCL1 promoter (*n* = 3). **p** SRC inhibitor Bosutinib (24 h) suppressed ETS2, p-ETS2 expression, and CXCL1 secretion in DLD1-sgControl cells, without any effect in SLC25A22 knockout cells (*n* = 3). Each dot represents an independent sample. Data are shown as mean ± SD (**f, l, m, o, p**). Two-tailed one-way ANOVA (**f, l, p**). Two-tailed Student's *t* test (**m, o**). Source data are provided as a Source Data file.

were maintained under specific pathogen-free conditions at the animal facility of CUHK. These mice were maintained in 12 h light/dark cycle, and the housing temperature and humidity were at 23 degrees and 45%, respectively. The maximal tumor size/burden (2 cm³) permitted by ethics committee was not exceeded in the study. All experimental procedures were approved by the Animal Ethics Committee of Chinese University of Hong Kong.

### PBMC humanized NSG mice
Human PBMC cells (STEMCELL Technologies) were implanted into NOD-SCID-γ (NSG) mice to generate PBMC humanized mice. Briefly, 4-weeks old NSG mice were intravenously injected $10^7$ PBMC cells 48 h after subcutaneous implantation of DLD1-SLC-KO or DLD1-sgControl cells ($2 \times 10^6$ cells/tumor). Human CD45$^+$ cells in the peripheral blood were determined by flow cytometry and mice with >25% hCD45$^+$ cells were considered humanized. All experimental procedures were approved by the Animal Experimentation Ethics Committee of CUHK.

### Western blot
Total proteins were extracted with CytoBuster reagent (Merck) containing protease and phosphatase inhibitor (Roche). Nuclear and cytoplasmic proteins were extracted with Nuclear and Cytoplasmic Extraction reagent (ThermoFisher). For western blot, proteins (10–30 µg) were separated on 10% SDS-PAGE and transferred onto a PVDF membrane. After blocking in 5% BSA in TBS-T (150 mmol/L Tris-HCl, pH 7.4, 50 mmol/L NaCl, 0.1% Tween 20), membranes were incubated with primary antibodies overnight at 4 °C, followed by secondary antibodies for 1 h at room temperature. Bands were visualized with Clarity Western ECL (Bio-Rad) in ChemiDoc XRS+ (Bio-Rad) using Image Lab software v6.1.0 (Bio-rad). The antibody list was shown in Table S5.

### Chromatin immunoprecipitation (ChIP)-PCR
Cells ($2 \times 10^7$) grown in a 15 cm dish were crosslinked with 1% formaldehyde for 10 min and quenched with 125 mM Glycine. Cell pellets were collected and lysed in SDS lysis solution (1% SDS, 10 mM EDTA in 5 mM Tris, pH8.1) containing protease inhibitors (Roche) for 30 min at 4 °C. Lysates were fragmented by ultrasonication (30 s on, 30 s off, 15 cycles) into 150-400 bp DNA, and immunoprecipitated with anti-ETS2 (5 µg, #PA5-28053, Thermo Fisher) or rabbit IgG (5 µg, sc-2027, Santa Cruz Biotechnology) overnight at 4 °C. Magna ChIP™ Protein A + G Magnetic Beads (Sigma-Aldrich) was added for 4 h, followed by washing and elution. Immunoprecipitated DNA were reverse crosslinked, purified, and quantified by qPCR. Primer sequences are listed in Table S1.

### Dual luciferase assay
ETS2 binding motif sequences on CXCL1 promoter were cloned into pGL3 luciferase reporter plasmid (Promega). Empty vector or ETS2 reporter were transfected into cells using FuGENE® HD (Promega). pRLTK Renilla Luciferase vector was co-transfected as internal control.

After 24 h, cells were harvested, and Firefly and Renilla luciferase activities were measured with Dual-Luciferase Reporter Assay System (Promega).

### Computational analysis of MDSC
MDSC signature (39 genes) was defined as described[16]. TCGA CRC (*N* = 383) were clustered into MDSC-low and MDSC-high group (distance between pairs of samples measured by Manhattan distance and clustering using complete-linkage hierarchical clustering method). Expression of SLC25A22 between MDSC-low and MDSC-high group were compared using Mann–Whitney test. Correlations between SLC25A22 and CXCL1/3 in human CRC tissues were analyzed using TCGA CRC dataset.

### Reverse transcription-quantitative polymerase chain reaction (RT-qPCR)
Total RNA was extracted from cells using Trizol Reagent (Thermo-Fisher). RNA purity and concentration were measured by ND-1000 (NanoDrop Technologies). RNA (1 µg) was reverse transcribed using PrimeScript Reverse Transcription Master Mix (TaKaRa, #RR036A). qPCR was performed with SYBR Green (Takara) in a QuantStudio Flex 7 Real Time PCR system (Quantstudio Real Time PCR software v1.7.2, Applied Biosystems) as follows: 10 min at 95 °C, and then 40 cycles of 15 s at 95 °C, 30 s at 60 °C and 30 s at 72 °C. All reactions were run in triplicates and normalized to GAPDH by $2^{-\Delta\Delta CT}$ method. Primer sequences are listed in Table S3.

### CRISPR-Cas9 knockout of SLC25A22
CRISPR-Cas9-mediated knockout was performed with lentiCRISPRv2 system. Briefly, guide RNAs targeting the murine genomic locus of SLC25A22 (sgSLC25A22 #1 and #2) were obtained from GenScript and cloned into the lentiCRISPRv2 vector (Addgene #108100). Lentivirus were then generated by co-transfection of lenti-CRISPRv2 vector and packaging plasmids in 293T cells. To perform knockout, $1 \times 10^5$ cells were infected with CRISPR-Cas9-gRNA lentivirus and selected by puromycin for 7 days. Puromycin-resistant clones were isolated and confirmed by western blot. sgRNAs for SLC25A22 were listed in Table S4.

### PCR array
Inflammatory Response and Autoimmunity PCR array (Qiagen) was used to examine the expression of 84 key genes involved in immune responses in SLC25A22 knockout cells according to the manufacturer's protocol. Data were normalized using the average of housekeeping genes on the same array and expression calculated by $2^{-\Delta\Delta CT}$ method.

### Antibody array
Cells ($1 \times 10^6$) were seeded into T75 flasks. Conditioned medium was collected at 24 h, and analyzed with Human cytokine/chemokine antibody array membranes (ab169812, Abcam) following the manufacturer's

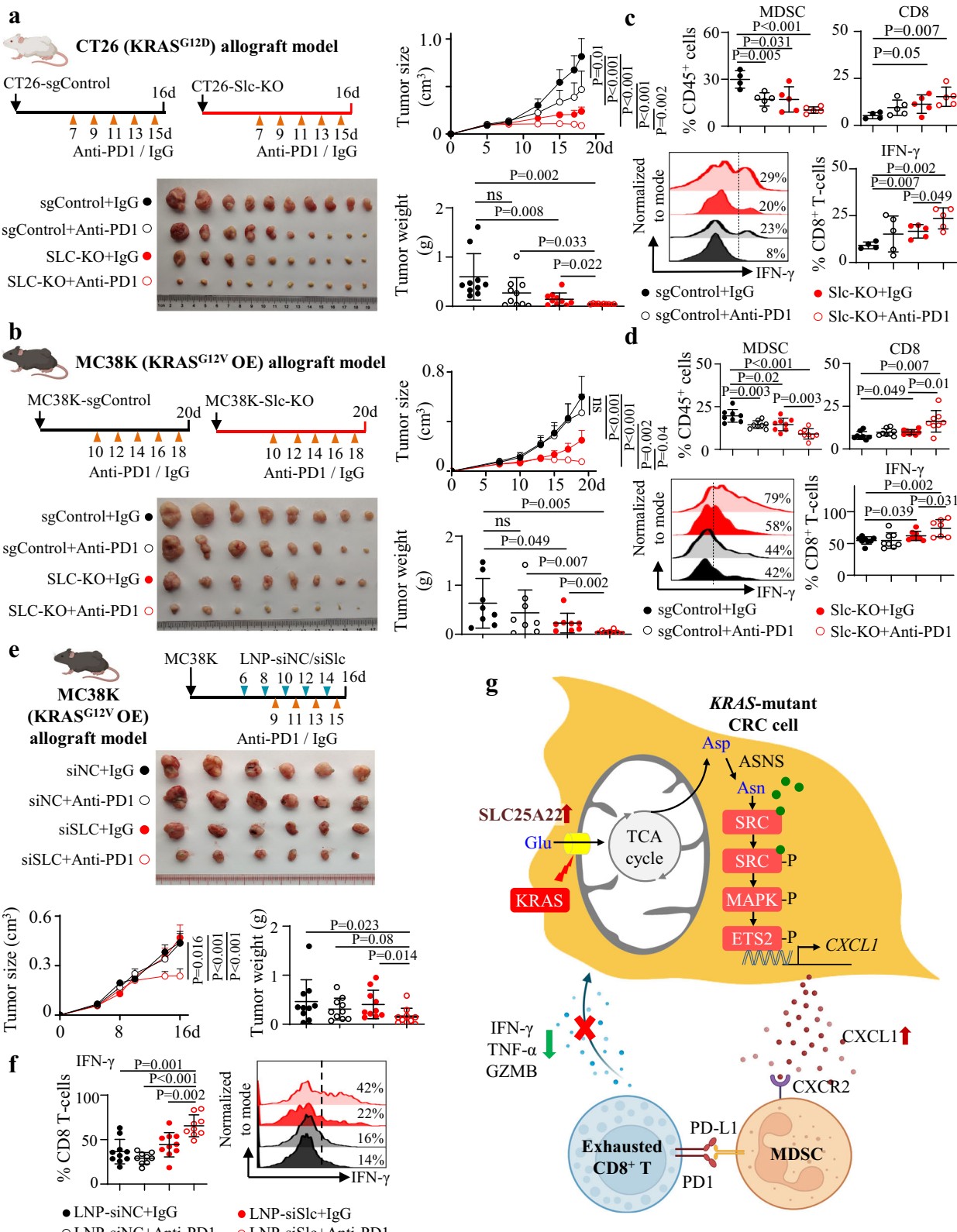

**Fig. 8 | SLC25A22 depletion synergized with anti-PD1 therapy in KRAS-mutant CRC. a** SLC25A22 knockout synergized with anti-PD1 in suppressing the growth of CT26 allografts ($n = 10$) and **b** MC38-Kras$^{G12V}$ allografts ($n = 8$). Each dot represents an independent tumor. **c** CD8$^+$ T-cell and IFN-γ expression was induced by combined SLC25A22 knockout and anti-PD1, whilst MDSC infiltration was reduced by the combination of SLC25A22 knockout plus anti-PD1 therapy in CT26 allografts ($n = 5$) and **d** MC38-Kras$^{G12V}$ allografts ($n = 8$). Each dot represents an independent tumor. **e** siSlc25a22-encapsulated nanoparticles (VNP-siSLC25A22) synergized with anti-PD1 to suppress MC38-Kras$^{G12V}$ allografts growth and increased CD8$^+$ T-cell activation

($n = 10$) and **f** their effect on IFN-γ levels on CD8$^+$ T cells (siNC+IgG and siSLC+IgG: $n = 10$ per group; siNC+Anti-PD1 and siSLC+ Anti-PD1: $n = 9$ per group). Each dot represents an independent tumor. **g** The overall graphical summary of the study (Created with BioRender.com). Data are presented as mean ± SEM for growth curve (**a**, **b**, **e**) and ± SD for others (**a**–**f**). Two-tailed two-way ANOVA for growth curve comparison (**a**, **b**, **e**). Two-tailed Student's *t* test for two-group comparison (**a**, **c**). Two-tailed Mann–Whitney *U* test for two-group comparison (**b**, **d**–**f**). ns, no significance. Source data are provided as a Source Data file.

protocol. For Human Phospho-Kinase Array Kit (ARY003C, R&D Systems), cells ($1 \times 10^6$) were cultured in 0.1 mM glutamine DMEM with FBS for 24 h and sequentially supplemented with 2 mM asparagine. The cells were harvested at the indicated time and analyzed. Density values of each spot measured in ImageLab software (Bio-Rad) and normalized by four positive control signals on the same membrane.

## Metabolomic analysis

Cells ($3 \times 10^6$) were cultured in glutamine-free DMEM supplemented with dialyzed FBS and 2 mM [U-$^{13}$C$_5$]-glutamine for indicated time points. Rinsed cells were treated with acetonitrile-water-formic acid (80:19:1, v/v/v) containing internal standard (4-Cl-Phe 0.5ppm) followed by freeze-thaw cycles using liquid N$_2$. Supernatants were dried under vacuum, reconstituted and analyzed by liquid chromatography-mass spectrometry.

## SLC25A22 knockout mice

Apc$^{min/+}$Kras$^{G12D}$ and Apc$^{min/+}$Kras$^{G12D}$Slc25a22$^{-/-}$ mice were generated as described[9]. Briefly, to generate Apc$^{min/+}$Kras$^{G12D}$ mice, Apc$^{min/+}$ mice were bred to Kras$^{G12D/+}$ and Villin-Cre mice to give Apc$^{min/+}$Kras$^{G12D/+}$Villin-Cre mice, which activated Kras$^{G12D}$ in intestinal tract. To achieve intestinal-specific SLC25A22 knockout in Apc$^{min/+}$Kras$^{G12D}$ mice, SLC25A22$^{fl/+}$ mice were bred to Apc$^{min/+}$Kras$^{G12D/+}$ and Villin-Cre mice, to give Apc$^{min/+}$ Kras$^{G12D/+}$SLC25A22$^{fl/+}$ and SLC25A22$^{fl/+}$Villin-Cre mice. Crossbreeding these gave Apc$^{min/+}$Kras$^{G12D/+}$SLC25A22$^{fl/fl}$Villin-Cre mice.

## Murine organoid isolation and culture

Apc$^{min/+}$Kras$^{G12D}$Villin-Cre and Apc$^{min}$Kras$^{G12D}$Slc25a22$^{fl/fl}$Villin-Cre mice organoids were established previously[9], and maintained in Matrigel with DMEM/F12 plus 10 mM HEPES, 2 mM Glutamax, N2 and B27 supplement, 1 mM N-acetylcysteine, 50 ng/mL murine epidermal growth factor, and penicillin-streptomycin.

## siRNA transfection

siRNA transfection was performed using Lipofectamine RNAiMAX reagent (Thermo Fisher) according to the manufacturers' protocol. Cells were transfected with siRNA for 48–72 h and harvested for qPCR and Western blot. For ELISA, 48 h post-transfection an equal number of cells were re-seeded and conditional medium were collected after 24 h and sterile filtered. siRNA sequences are listed in the Table S6.

## siRNA-loaded nanoparticles preparation and in vivo treatment

siRNA-loaded nanoparticles were manufactured by a double emulsion method. Briefly, an aqueous solution of siRNA (15 nmol, 25 µL) was emulsified by sonication on ice for 1 min in chloroform (0.5 mL) with cationic lipid DOTAP (1 mg) and mPEG5k-b-PLGA11k (25 mg). Primary emulsion was further emulsified in 5 mL of DEPC water by sonication (65 W, 2 min) on ice. Next, chloroform was removed using rotary evaporator. The obtained siRNA-loaded nanoparticles were stored at 4 °C. VNP-siNC/siSLC25A22 (0.5 OD per mice for intratumoral injection) treatment was initiated when tumor volume reached an average of ~100 mm$^3$ at a frequency of every 2 days. Anti-PD1 (200 µg per mice, i.p., BE0146, Bio X Cell) started after 2 doses of siRNA at a frequency of every 2 days. All experimental procedures were approved by the Animal Ethics Committee of Chinese University of Hong Kong.

## SRC kinase activity assay

The effect of asparagine on SRC activity were measured using SRC Kinase Enzyme System (V9741, Promega). Briefly, 2 ng recombinant SRC protein were pre-incubated with or without asparagine at 37 °C for 30 min in Kinase Buffer (40 mM Tris, pH 7.5; 20 mM MgCl$_2$; 0.1 mg/ml BSA; 2 mM MnCl$_2$; 50 µM DTT), followed by addition of SRC substrate (0.2 µg/µl) and ATP (500 µM) for 60 min. Luciferase detection was performed according to the manufacturer's instructions.

## Statistical analysis

Data are presented as mean ± SD or mean ± SEM. Statistical analysis was performed using Prism 8.0 (GraphPad Software, Inc.). Differences between two groups were compared with Mann–Whitney $U$ test or Student's $t$ test. Multiple group comparisons were conducted using one-way analysis of variance (ANOVA) followed by multiple comparisons. A two-way analysis of variance was used to compare growth curves. Correlation was performed using Pearson Correlation test and Spearman Correlation test. $P < 0.05$ was considered as statistically significant.

## Reporting summary

Further information on research design is available in the Nature Portfolio Reporting Summary linked to this article.

## Data availability

All TCGA dataset was accessed using UCSC Xena Browser (https://xenabrowser.net/). We have downloaded normalized RNA-seq datasets directly from Xena Browser from TCGA Colon and Rectal Cancer. Raw data from targeted metabolite analysis is available upon request to the corresponding author. The remaining data are available within the Article, Supplementary Information, or Source Data file. Source data are provided in this paper.

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

## Acknowledgements

This project was supported by funds from Research Grants Council-General Research Fund (24100520, 14101917, 14101321), Research Grants Council-Collaborative Research Fund (C4039-19G); Heath and Medical Research Fund (06170686, 08190706); and Science and Technology Program Grant Shenzhen (JCYJ20170413161534162). We thank Minnie Y.Y. Go for support with experiments. Images in Figs. 1d, 2a, 4a, 5a and 8a, b, e, g were created with BioRender.com.

## Author contributions

Q.Z. performed the experiments, analyzed the data, and drafted the manuscript. Y.P., H.C., L.S.C., H.G., P.H., D.C., and Q.W. assisted in the experiments. F.J. performed LC-MS analysis. W.K. performed a histological assay. H.S., C.L., and X.Z. assisted in mice models. Y.L. performed bioinformatic analysis. J.Y. and C.C.W. designed and supervised the project and revised the manuscript.

## Competing interests

The authors declare no competing interests.
