## [Peer Review File · Nature Communications]

Targeting of SLC25A22 boosts the immunotherapeutic response in KRAS-mutant colorectal cancerReviewers' comments:

Reviewer #1 (Remarks to the Author): with expertise in cancer immunology, MDSC

In this manuscript, the authors showed SLC25A22 in mutant KRAS induced immune suppression in CRC. Mechanistically, we found that SLC25A22 plays a central role in promoting asparagine, which binds and activates SRC phosphorylation. Asparagine mediated SRC promotes ERK/ETS2 signaling, latter which drives CXCL1 transcription. Secreted CXCL1 functions as a chemoattractant for MDSC via CXCR2, leading to an immunosuppressive microenvironment. However, several critical points should be addressed to make this study suitable for publication.

Major Points:

- Background on SLC25A22 has not been developed in the introduction. The authors should provide with more exhaustive information about this transporter.
- The authors claim that RNA-seq revealed cytokine-cytokine receptor as the top enriched pathway in tumors compared to adjacent normal tissues in APC-KRAS mutant mice (Fig. 1B), indicating altered tumor immunity. Based on cytokine-cytokine receptor changes in CRC, why do the authors make the conclusion that there is an altered tumor immunity? Clarify this point.
- The authors compared the immune landscape of tumors from APC-KRAS mutant mice to normal adjacent tissue and they find higher frequency of MDSC by CYTOF and flow cytometry. It is not surprising to find MDSC in tumors compared to normal tissue, however the authors did not compare immune landscape of WT and APC mutant. Clarify this critical point because the authors claim that KRAS-mutation promotes immunosuppression.
- SLC25A22 loss abolishes mutant KRAS-induced immunosuppression in APC KRAS mutant organoids and CT26 allograft models. The authors show that APC-KRAS-SLC25A22KO tumors exhibited arrested growth compared to APC-KRAS mutant tumors, as well as reduction in the frequency of MDSC. They showed that secreted CXCL1 functions as a chemoattractant for MDSC via CXCR2, in a SLC25A22 dependent manner. The frequency of MDSC in tumors depends on the tumor size and increases overtime during cancer progression. Is it possible that reduction in MDSC in tumors may depend on the fact that SLC25A22 affects the proliferation of tumor cells rather than the recruitment of MDSC? The authors should clarify the ability of SLC25A22 to affect proliferation of tumor cells and to verify the levels of CXCL1 in WT versus KO SLC25A22.
- The authors showed that CD33+ cells increased in human CRC. CD33 a specific myeloid marker, is CD33 sufficient to make the conclusion that CD33+ cells are truly MDSC? Clarify this critical point. Moreover, the presence of CD33+ cells positively correlated with SLC25A22 in KRAS mutant CRC. It is unclear whether there is difference in the expression of SLC25A22 in KRAS mutant CRC versus KRAS WT CRC. The authors claimed that SLC25A22 underlies mutant KRAS34 induced immune suppression in CRC. Thus, I would expect that SLC25A22 expression is much higher in KRAS mutant CRC versus KRAS WT CRC. Clarify this critical point.
- In mouse model KRAS mutant mice do not develop tumors, contrary to KRAS-APC mutants that develop colon tumors. Thus, what are the characteristics of human samples analyzed in the study?

Reviewer #2 (Remarks to the Author): with expertise in cancer therapy, asparaginase

In this manuscript, Zhou et al work to investigate the role of SLC25A22 (best known as a mitochondrial glutamate transporter) as a repressor of antitumor immunity in KRAS-mutant CRC. The authors link this effect to a complicated mechanism involving asparagine transport, SRC binding and activation, ERK upregulation, CXCL1 transcription and secretion, and its ability to attract MDSC resulting in an immunosuppressive microenvironment. Unfortunately, while several parts of this manuscript are promising, I am afraid this manuscript attempts to prove too many points, without convincingly proving many of them before moving on to the next similarly problematic point/conclusion. Two examples are provided here:

1. In the first paragraph of the results and the first part of Figure 1, the conclusion the authors aim to prove is that "KRAS mutation promotes immunosuppression" (line 94). However, these data are all

from comparisons of tumor vs normal tissue in APC-KRAS mutant mice; isn't KRAS expressed in both normal and tumor tissue in these mice? Doesn't this mean that any difference between these cells is NOT due to KRAS mutations? I do not understand how one can conclude that KRAS is promoting immunosuppression without comparing KRAS mutant vs KRAS wild-type tumors, which is not what was performed.

2. In the second paragraph of the results, the conclusion the authors aim to prove is that SLC25A22 loss abolishes KRAS-induced immunosuppression. However, the experiment is deleting Slc25a22 in CRC, implanting these into immunocompetent mice, and showing that these impair tumor growth and immune tumor infiltrates. Couldn't these findings be explained if SLC25A22 loss is simply toxic to CRC cells, without any direct effects on tumor immune infiltration? The fact that Slc25a22 loss is toxic to CRC was shown by the authors in a prior paper (ref #8 cited by the authors)?

Unfortunately, these and many other issues rise to the level of fatal flaws in this manuscript. I recommend that the authors focus much more narrowly on a small set of conclusions to be proven, and that each one be proven with multiple orthogonal and well-controlled experimental approaches.

Reviewer #3 (Remarks to the Author): with expertise in colorectal cancer, immunotherapy

Zhou et al characterize a novel mechanism of tumor microenvironment immunosuppression in KRAS mutant colorectal cancer by which the mitochondrial transporter SLC2A22 drives CXCL1 expression, leading to increased myeloid derived suppressor cell (MDSC) recruitment and decreased cytotoxic CD8 T cell infiltration. The authors rigorously explored the mediators of CXCL1 expression downstream of SLC2A22, implicating glutamine and Asparagine metabolism as well as SRC kinase in this axis. They show that inhibition of this SLC2A22 axis leads to decreased MDSC infiltration and increased T cell infiltration, as well as decreased MDSC immunosuppressive capacity and increased T cell cytotoxicity in vitro. They finally show that inhibition of SLC2A22 via siRNA nanoparticles synergizes with anti PD1 immunotherapy to inhibit mouse allograft tumor growth, highlighting SLC2A22 as a potential therapeutic target that may increase CRC response rates to checkpoint immunotherapy. Overall, this is an interesting and well written manuscript, with experiments generally performed to a high standard. The inclusion of a PBMC humanized mouse model for some experiments is a welcome touch. The manuscript uses a wide array of human and mouse models, with different models used for different experiments, which is somewhat distracting and not all conclusions are consistent across models. The final figure of the manuscript ultimately arguing for the clinical translatability of inhibiting the SLC2A22 axis in combination with checkpoint immunotherapy uses MC38 (MSI-H) and CT26 (an unusual, IO responsive chemical carcinogen induced mouse CRC cell line) mouse cell line allograft models, thus the relevance of the immunomodulatory biology uncovered to advanced CRC in patients with MSS KRAS mutant tumors (the vast majority of KRAS mutant CRC) is unclear. Nonetheless, the authors elegantly uncovered a mechanism of immunosuppression driven by a novel metabolic pathway, and showed how targeting this pathway in CRC can potentially synergize with anti-PD1 to increase responses to checkpoint immunotherapy, something there is indeed a significant clinical need for. Their data supports further interrogation of these pathways in more advanced models of the human tumor immune microenvironment.

Major points

- Clinical data are restricted to scoring of immune subpopulations in TMAs from CRC patients. MSI status of the patients is not reported. While there are some published data supporting worse prognosis in patients with MSI-H CRC whose tumors also harbor KRAS mutations, can the authors comment on whether KRAS mutation has been associated with worse response to immunotherapy?
- Regardless, the major question remains whether targeting SLC25A22 is a credible approach in MSS CRC, which is largely unresponsive to current checkpoint immunotherapy regardless of KRAS mutation status. The GEMM model used here only generates (small intestinal) adenomas, not colorectal cancer. The authors should introduce TP53 mutations into the murine organoids with

APC/KRAS +/- SLC25A22 mutations (in the form used only a model of adenoma, not adenocarcinoma) and perform orthotopic transplantation, tumor growth and PD-1 checkpoint inhibition assays to ascertain the relevance of the biology described in an immunocompetent, MSS, immune cold adenocarcinoma context that is more applicable to human colorectal cancer.

- While discussing the data in Figure 8, the authors state that in CT26 allografts and MC38-KRASG12V allografts, anti PD1 alone had no significant effect on tumor size. However, in Figure 8A and 8B in the photos of resected tumors from each group, the tumors in the anti PD1 alone group (3rd from top) are quite noticeably smaller than the sgControl + IgG group (top row). In 8B, the tumor size graph on the right shows that sgControl + IgG and sgControl + Anti-PD1 are essentially identical in size, which is simply not supported by the photo on the left. Additionally, the authors state that while anti PD1 alone had no significant effect on tumor size, SLC-KO + IgG had a 50% reduction in size. In both 8A and 8B, the anti PD1 alone tumors (3rd from the top) appear at least similar in size to the SLC-KO+IgG tumors (2nd from top), if not slightly smaller, suggesting that anti-PD1 caused a similar percent reduction in volume as SLC-KO + IgG. This completely contradicts the tumor size graph on the right that shows the sgControl + Anti PD1 tumors are larger than the SLC-KO + IgG tumors. In the photos in both 8A and 8B, the tumors in the top row visually appear largest, and the size decreases with each row from top to bottom. If the photos on the left were in fact accidentally mislabeled, and the 2nd from the top row is the sgControl + AntiPD1 group and the 3rd from the top row is SLC-KO + IgG group, the photos would be much more consistent with the graphs on the right. While the synergy of SLC-KO with anti-PD1 is obvious from the smallest tumors in the bottom row of the photos which is ultimately the conclusion of this experiment, the authors should clarify these discrepancies between the photos of the tumors and the graphs. If it was the case that a simple mislabeling occurred, the authors should still clarify if the differences in size and weight between sgControl + IgG and sgControl + Anti-PD1 in 8A are in fact nonsignificant with an “ns” label or ideally the P value, as there are no statistics shown currently for that comparison.

- In Figure 5, the authors functionally characterize MDSCs and T cells, the two main immune populations of interest in this paper. They perform coculture experiments of MDSCs and T cells to demonstrate MDSC immunosuppressive capacity, and interrogate the activation status and cytotoxicity of T cells by staining for IFN γ , TNF α , and Granzyme B. All of these data are very convincing. This validation of immune cell function is completely absent from earlier figures, where the authors simply quantify these immune populations and their ratio to make statements about whether a tumor is immunosuppressed. For example, the authors show in Figure 1 that SLC25A22 knockout in APC KRAS mouse tumors leads to decreased MDSC abundance and increased T cell abundance, and they conclude that SLC25A22 knockout reverses KRAS-induced immunosuppression. Without any functional characterization of the activation status of these T cells or their tumor killing capacity *ex vivo*, no conclusions can be drawn about the immunosuppressive nature of these tumors. All that can be said is that the knockdown altered the T cell abundance. It could be possible that the T cells are more abundant but just as exhausted and dysfunctional and no less suppressed than T cells in tumors without SLC25A22 knockout. The authors should ideally do these functional experiments, or at least alter their language used when describing immune cell infiltration in their models to be less definitive.

Additional points

- In figure 2C where the authors show their gating strategy to quantify MDSCs and T cells, it's not clear from the density plots shown how the gates were drawn and how the cutoffs were determined. For the CD4 vs CD8 pool in particular, the CD4+ positive and negative populations are not obvious, which can make the gate look somewhat arbitrarily drawn. It would be helpful if the authors overlaid the unstained control sample on these plots to show how the gates were drawn.

- Given that response to checkpoint immunotherapy in colorectal cancer varies dramatically based on tumor mutation burden, microsatellite instability, and mismatch repair pathway status, it would be important to know this information about each mouse and human cell line used in this study in order to interpret the anti PD1 treatment data.

- In the discussion section, the authors primarily reiterate their results and the conclusions they drew from them. There is little to no discussion about where this data fits in the fields of cancer metabolism and cancer immunotherapy, either in relation to previous studies in this area or future directions of the fields. They cite and discuss a single paper from the literature as it related to their findings, Oh et al. 2020. Overall, the authors should put their findings into a larger context.

Reviewer #4 (Remarks to the Author): with expertise in cancer metabolic targeting

This manuscript presents a substantial set of experimental evidence that the mitochondrial glutamate transporter SLC25A22 regulates immunosuppression via CXCL1 secretion and recruitment of myeloid-derived suppressor (MDSC) cells in models of KRAS-mutant colorectal cancer (CRC). While glutamine/glutamate metabolism has been linked previously to immunosuppression in syngeneic CRC models (Leone et al. Science 2019), the proposed mechanistic significance of MDSC distinguishes the current work. There is some corroboration in human CRC datasets/cell lines. Another apparently novel aspect of the work is the assertion that the regulation of Src activity by Asn links glutamate metabolism to CXCL1 production in CRC cells. The work is significant in that it potentially offers new routes to reverse the immunosuppressive phenotype of KRAS-driven CRC, a subtype which remains difficult to target.

The conclusions are for the most part well supported by the work shown. The most confusing aspect for me relates to the key experiments that first identify Asn as the one downstream metabolite that rescues CXCL1/3 expression/secretion in SLC25A22 deficient cells (Panels 6F/G). I cannot understand, if metabolic production of Asn from glutamate is key, why e.g. extracellular aspartate does not also rescue the phenotype. This is particularly so in light of the subsequent panels where it is shown that interference with the enzyme ASNS (which converts aspartate to asparagine) also inhibits CXCL1 expression – so clearly these cell lines possess sufficient capacity to convert aspartate in the first place. This suggests that some additional experiments may be necessary.

- It could be the case that adding extracellular Asn boosts Src activity and CXCL1/3 production regardless of SLC25A22 status (which would also imply that while losing metabolic uptake glu does reduce Asn it does not fully explain why cytokine production is reduced). The experiments in panel 6F should include what happens to the sgControl and both SLC-KO1/2 in both backgrounds (DLD1, CT26) and perhaps also the parental lines in supplementary. Similarly Panel 6G should also include what happens to the sgControl lines. Also more cell permeable versions of a-KG, succinate etc (e.g. the methyl esters) should be used to see if these rescue. The concentrations of the added metabolites should be given in legend/methods.
- With respect to the experiments above using Asn should be repeated with a titration to the extracellular concentration required to achieve the minimum (low uM) levels in the cell that are required to active Src (0.6uM-6uM, Figure 7F). If this intracellular Asn level does not rescue CXCL1/3 expression then it is less likely that Src activation is the link between SLC25A44 and cytokine expression.
- It should be proven that the addition of aspartate or cell permeable metabolites do not rescue intracellular Asn concentrations to the level of sgControl and SLC-KO1/2 in both lines (Fig 6H) – if they do, but do not rescue cytokine expression then again it seems unlikely that the link to SLC25A44 mediation of CXCL1/3 holds. The conversion or not of ¹³C labelled aspartate should be reported in the sgControl and SLC-KO lines, and also in the siASNS lines to prove functional interference. The knockdown efficiencies of siASNS in the timeframe of the experiments (in Fig 6J/K) should also be shown with western blot.
- I do not understand in Fig 6J/K when using asparaginase, how in the absence of Asn in the DMEM media the treatment can have an effect i.e. how does it penetrate the cell and lower intracellular Asn to effect Src etc? That does not seem plausible, but metabolomics could be used to prove the effect on intracellular Asn.
- Figures 7D, E & F – D-asparagine should be used as a control for these experiments to show specificity for the biologically relevant enantiomer and this could be a relevant control for some the experiments above.
- The implication of this paper is in effect that SLC25A44 regulates MDSC via ASNS and Src. Surely then more established therapeutics such as Asparaginase and Src inhibitors would induce the same effects as targeting SLC25A44? Providing experimental evidence for this in vivo would be important to prove the authors' hypothesis and these key experiments would significantly enhance the

translatability of their findings to the clinic.

In the discussion it is argued that there would be a therapeutic window for targeting SLC25A44 because other immune cells can use extracellular Asn. However the authors' own data shows that extracellular Asn can rescue immunosuppressive factor secretion in SLC25A44 negative cells, so how would this approach work in vivo? This weakness should be acknowledged in discussion.

Response to Reviewer #1:

In this manuscript, the authors showed SLC25A22 in mutant KRAS induced immune suppression in CRC. Mechanistically, we found that SLC25A22 plays a central role in promoting asparagine, which binds and activates SRC phosphorylation. Asparagine mediated SRC promotes ERK/ETS2 signaling, latter which drives CXCL1 transcription. Secreted CXCL1 functions as a chemoattractant for MDSC via CXCR2, leading to an immunosuppressive microenvironment. However, several critical points should be addressed to make this study suitable for publication.

Major Points:

1. Background on SLC25A22 has not been developed in the introduction. The authors should provide with more exhaustive information about this transporter.

Response: We have now added new information as follows: “Our previous work has shown that SLC25A22^{8,9}, a member of mitochondrial transporter family (SLC25) that facilitates the transport of glutamate across the inner mitochondrial membrane into the mitochondrial matrix^{10,11}, functions as an oncogenic factor in KRAS mutant CRC by generating metabolites critical for antioxidative defense⁸ and epigenetic deregulation⁹” on **p.3, line 70-74**.

2. The authors claim that RNA-seq revealed cytokine-cytokine receptor as the top enriched pathway in tumors compared to adjacent normal tissues in APC-KRAS mutant mice (Fig. 1B), indicating altered tumor immunity. Based on cytokine-cytokine receptor changes in CRC, why do the authors make the conclusion that there is an altered tumor immunity? Clarify this point.

Response: We have now compared transcriptomes of colon tumors of APC- and APC-KRAS mutant mice (**Fig. 1B**), showing that multiple pathway associated with immune response are enriched in APC-KRAS mutant mouse tumors, such as TNF and Cytokine-cytokine receptor pathways. We thus hypothesized that mutant KRAS might modulate immune landscape, which we further validated using flow cytometry (**Fig. 1C**). This information has been added to **p.4, line 95-99**.

3. The authors compared the immune landscape of tumors from APC-KRAS mutant mice to normal adjacent tissue and they find higher frequency of MDSC by CYTOF and flow cytometry. It is not surprising to find MDSC in tumors compared to normal tissue, however the authors did not compare immune landscape of WT and APC mutant. Clarify this critical point because the authors claim that KRAS-mutation promotes immunosuppression.

Response: To clarify this point, we have now performed analysis of CRC tumors from $Apc^{Min/+}$ and $Apc^{Min/+}Kras^{G12D/+}$ mice by flow cytometry (**Fig. 1C**). We demonstrated that $Apc^{Min/+}Kras^{G12D/+}$ tumors had much higher infiltration of total MDSC ($P<0.001$) and PMN-MDSC ($P<0.001$) compared to $Apc^{Min/+}$ tumors. On the contrary, total T-cells ($P<0.01$) and CD8⁺ T-cells ($P<0.001$) were decreased in $Apc^{Min/+}Kras^{G12D/+}$ compared to $Apc^{Min/+}$ tumors. These findings are consistent with the notion that mutant KRAS promotes immunosuppression. This information has been added to **p.4, line 99-102**.

4. SLC25A22 loss abolishes mutant KRAS-induced immunosuppression in APC KRAS mutant organoids and CT26 allograft models. The authors show that APC-KRAS-SLC25A22-KO tumors exhibited arrested growth compared to APC-KRAS mutant tumors, as well as reduction in the frequency of MDSC. They showed that secreted CXCL1 functions as a chemoattractant for MDSC via CXCR2, in a SL25A22 dependent

manner. The frequency of MDSC in tumors depends on the tumor size and increases overtime during cancer progression. Is it possible that reduction in MDSC in tumors may depend on the fact that SLC25A22 affects the proliferation of tumor cells rather than the recruitment of MDSC? The authors should clarify the ability of SLC25A22 to affect proliferation of tumor cells and to verify the levels of CXCL1 in WT versus KO SLC25A22.

Response: To clarify this point, we have now performed MTT assay and CXCL1 assay in the APC-KRAS mutant organoids. MTT assay showed that SLC25A22 knockout did suppress cell viability (**Fig. S5**). Nevertheless, even when normalized to cell viability, CXCL1 secretion in APC-KRAS mutant organoids with SLC25A22-KO was impaired compared to WT (**Fig. S5**), suggesting that SLC25A22 regulates CXCL1 independently of cell proliferation. This suggests that SLC25A22 knockout, at least in part, suppress tumor growth via regulation of MDSC recruitment. This information has been added to **p.6, line 184-186**.

5. The authors showed that CD33+ cells increased in human CRC. CD33 a specific myeloid marker, is CD33 sufficient to make the conclusion that CD33+ cells are truly MDSC? Clarify this critical point. Moreover, the presence of CD33+ cells positively correlated with SLC25A22 in KRAS mutant CRC. It is unclear whether there is difference in the expression of SLC25A22 in KRAS mutant CRC versus KRAS WT CRC. They authors claimed that SLC25A22 underlies mutant KRAS induced immune suppression in CRC. Thus, I would expect that SLC25A22 expression is much higher in KRAS mutant CRC versus KRAS WT CRC. Clarify this critical point.

Response: CD33 is an established marker for MDSCs in humans, and has been applied for the immunostaining for MDSCs other publications (Taki, et al, *Nat Commun*, 2018, 9, 1685; Man, et al, *Gut*, 2020, 69, 365-379). These references have been added to the main text on **p.5, line 146**.

Our previous research in *in vitro* cell cultures using isogenic colon cell lines revealed that mutant KRAS promotes SLC25A22 expression (Wong CC, et al. *Gastroenterology*, 2020). In TMA cohort % SLC25A22-high (IHC score 3) tumors is higher in KRAS-mutant CRC (30/104, 28.8%) compared to KRAS wildtype CRC (24/103, 23.3%). We previously showed that SLC25A22 overexpressed in CRC only predicts poor survival in KRAS-mutant CRC patients, but not KRAS-wildtype CRC, suggesting it is more important in the context of mutant KRAS (Wong CC, et al. *Gastroenterology*, 2016).

6. In mouse model KRAS mutant mice do non-develop tumors, contrary to KRAS-APC mutants that develops colon tumors. Thus, what are the characteristics of human samples analyzed in the study?

Response: We have now added the APC status to the TMA cohort. A majority of CRC with mutant KRAS harboured co-mutations in APC (77/102) (75.4%), suggesting that APC-KRAS mutant mice model mimics mutational profile in humans. This has been added to **p.5, line 143-145**.

Response to Reviewer #2:

1. In the first paragraph of the results and the first part of Figure 1, the conclusion the authors aim to prove is that “KRAS mutation promotes immunosuppression” (line 94). However, these data are all from comparisons of tumor vs normal tissue in APC-KRAS mutant mice; isn't KRAS expressed in both normal and tumor tissue in these mice? Doesn't this mean that any difference between these cells is NOT due to KRAS mutations? I do not understand how one can conclude that KRAS is promoting immunosuppression without comparing KRAS mutant vs KRAS wild-type tumors, which is not what was performed.

Response: To clarify this point, we have now analyzed CRC tumors from *Apc*^{Min/+} and *Apc*^{Min/+}*Kras*^{G12D/+} mice by RNA-sequencing and flow cytometry. RNA-seq showed that multiple pathway associated with immune response are enriched in APC-KRAS mutant mice, such as TNF and Cytokine-cytokine receptor pathways, suggesting that mutant KRAS might alter the immune landscape (**Fig. 1B**). We have now performed analysis of CRC tumors from *Apc*^{Min/+} and *Apc*^{Min/+}*Kras*^{G12D/+} mice by flow cytometry (**Fig. 1C**). We demonstrated that *Apc*^{Min/+}*Kras*^{G12D/+} tumors had higher infiltration of total MDSC (P<0.001) and PMN-MDSC (P<0.001) compared to *Apc*^{Min/+} tumors. On the contrary, total T-cells (P<0.01) and CD8⁺ T-cells (P<0.001) were reduced. These results are consistent with the notion that mutant KRAS promotes immunosuppression. This information has been added to **p.4, line 95-102**.

2. In the second paragraph of the results, the conclusion the authors aim to prove is that SLC25A22 loss abolishes KRAS-induced immunosuppression. However, the experiment is deleting *Slc25a22* in CRC, implanting these into immunocompetent mice, and showing that these impair tumor growth and immune tumor infiltrates. Couldn't these findings be explained if SLC25A22 loss is simply toxic to CRC cells, without any direct effects on tumor immune infiltration? The fact that *Slc25a22* loss is toxic to CRC was shown by the authors in a prior paper (ref #8 cited by the authors)?

Response: We agree that we could not exclude the possibility that SLC25A22 have an effect on cell viability. Nevertheless, the effect of SLC25A22 on cytokines mRNA and secretion is still highly significant after adjusting for cell viability (**Fig. S5**). Moreover, anti-CD8 depletion experiment (**Fig. 5G**) demonstrated that the tumor promoting effect of SLC25A22 was dependent on functioning CD8⁺ T-cells. Taken together, the impact of knocking out SLC25A22 is, at least in part, mediated by its effect on tumor immunity.

Response to Reviewer #3:

Zhou et al characterize a novel mechanism of tumor microenvironment immunosuppression in KRAS mutant colorectal cancer by which the mitochondrial transporter SLC2A22 drives CXCL1 expression, leading to increased myeloid derived suppressor cell (MDSC) recruitment and decreased cytotoxic CD8 T cell infiltration. The authors rigorously explored the mediators of CXCL1 expression downstream of SLC2A22, implicating glutamine and Asparagine metabolism as well as SRC kinase in this axis. They show that inhibition of this SLC2A22 axis leads to decreased MDSC infiltration and increased T cell infiltration, as well as decreased MDSC immunosuppressive capacity and increased T cell cytotoxicity in vitro. They finally show that inhibition of SLC2A22 via siRNA nanoparticles synergizes with anti PD1 immunotherapy to inhibit mouse allograft tumor growth, highlighting SLC2A22 as a potential therapeutic target that may increase CRC response rates to checkpoint immunotherapy. Overall, this is an interesting and well written manuscript, with experiments generally performed to a high standard. The inclusion of a PBMC humanized mouse model for some experiments is a welcome touch. The manuscript uses a wide array of human and mouse models, with different models used for different experiments, which is somewhat detracting and not all conclusions are consistent across models. The final figure of the manuscript ultimately arguing for the clinical translatability of inhibiting the SLC2A22 axis in combination with checkpoint immunotherapy uses MC38 (MSI-H) and CT26 (an unusual, IO responsive chemical carcinogen induced mouse CRC cell line) mouse cell line allograft models, thus the relevance of the immunomodulatory biology uncovered to advanced CRC in patients with MSS KRAS mutant tumors (the vast majority of KRAS mutant CRC) is unclear. Nonetheless, the authors elegantly uncovered a mechanism of immunosuppression driven by a novel metabolic pathway, and showed how targeting this pathway in CRC can potentially synergize with anti-PD1 to increase responses to checkpoint immunotherapy, something there is indeed a significant clinical need for. Their data supports further interrogation of these pathways in more advanced models of the human tumor immune microenvironment.

Major points

1. Clinical data are restricted to scoring of immune subpopulations in TMAs from CRC patients. MSI status of the patients is not reported. While there are some published data supporting worse prognosis in patients with MSI-H CRC whose tumors also harbour KRAS mutations, can the authors comment on whether KRAS mutation has been associated with worse response to immunotherapy?

Response: In our TMA cohort, only a very minor portion of CRC patients had MSI-H status (13/203, 6.4%; KRAS-mutant: 7; KRAS-wildtype: 6). The low patient numbers preclude detailed survival analysis. Meanwhile, KRAS-mutation has been shown to be associated with non-response to PD-1 inhibitors in MSS CRC (Sun et al., *Frontiers in Oncology*, 2021, 11, 754881). We have added the following to the discussion section on **p.15, line 478-479**.

2. Regardless, the major question remains whether targeting SLC25A22 is a credible approach in MSS CRC, which is largely unresponsive to current checkpoint immunotherapy regardless of KRAS mutation status. The GEMM model used here only generates (small intestinal) adenomas, not colorectal cancer. The authors should introduce TP53 mutations into the murine organoids with APC/KRAS +/- SLC25A22 mutations (in the form used only a model of adenoma, not adenocarcinoma) and

perform orthotopic transplantation, tumor growth and PD-1 checkpoint inhibition assays to ascertain the relevance of the biology described in an immunocompetent, MSS, immune cold adenocarcinoma context that is more applicable to human colorectal cancer.

Response: To mimic TP53 dysfunction in CRC, we have depleted TP53 in APC-KRAS mutant and APC-KRAS-SLC25A22^{KO} organoids following the protocol of Onuma et al (PNAS, 2013, 110, 11127-11132), and then performed orthotopic injection and PD-1 therapy. *Apc*^{Min/+}*Kras*^{G12D/+}*shTp53* was resistant anti-PD1, while *Apc*^{Min/+}*Kras*^{G12D/+}*Slc25a22*^{-/-}*shTp53* in conjunction with anti-PD1 most effectively reduced the growth of orthotopic tumors (P<0.001), and was more effective than single anti-PD1 (P<0.01) or SLC25A22 knockout (P<0.05) treatment (**Fig. S14A-B**). Flow cytometry showed that combined SLC25A22 knockout plus anti-PD1 suppressed tumor infiltrating MDSC and PMN-MDSC, while CD8⁺ T-cells and IFN- γ ⁺ CD8⁺ T-cells were induced (**Fig. S14C**). Hence, targeting of SLC25A22 is effective in an immunocompetent, MSS, and immune cold adenocarcinoma context. This information has been added to **p.12, line 364-369**.

3. While discussing the data in Figure 8, the authors state that in CT26 allografts and MC38-KRASG12V allografts, anti PD1 alone had no significant effect on tumor size. However, in Figure 8A and 8B in the photos of resected tumors from each group, the tumors in the anti PD1 alone group (3rd from top) are quite noticeably smaller than the sgControl + IgG group (top row). In 8B, the tumor size graph on the right shows that sgControl + IgG and sgControl + Anti-PD1 are essentially identical in size, which is simply not supported by the photo on the left. Additionally, the authors state that while anti PD1 alone had no significant effect on tumor size, Slc-KO + IgG had a 50% reduction in size. In both 8A and 8B, the anti PD1 alone tumors (3rd from the top) appear at least similar in size to the SLC-KO+IgG tumors (2nd from top), if not slightly smaller, suggesting that anti-PD1 caused a similar percent reduction in volume as SLC-KO + IgG. This completely contradicts the tumor size graph on the right that shows the sgControl + Anti PD1 tumors are larger than the SLC-KO + IgG tumors. In the photos in both 8A and 8B, the tumors in the top row visually appear largest, and the size decreases with each row from top to bottom. If the photos on the left were in fact accidentally mislabeled, and the 2nd from the top row is the sgControl + AntiPD1 group and the 3rd from the top row is SLC-KO + IgG group, the photos would be much more consistent with the graphs on the right. While the synergy of SLC-KO with anti-PD1 is obvious from the smallest tumors in the bottom row of the photos which is ultimately the conclusion of this experiment, the authors should clarify these discrepancies between the photos of the tumors and the graphs. If it was the case that a simple mislabeling occurred, the authors should still clarify if the differences in size and weight between sgControl + IgG and sgControl + Anti-PD1 in 8A are in fact nonsignificant with an “ns” label or ideally the P value, as there are no statistics shown currently for that comparison.

Response: We apologize for the error, as labeling order of tumors between sgControl + Anti-PD1 and SLC-KO+IgG were indeed reversed in the tumor images. We have also checked all the statistically significance among all the groups and added as appropriate on **Fig. 8A and 8B**.

4. In Figure 5, the authors functionally characterize MDSCs and T cells, the two main immune populations of interest in this paper. They perform coculture experiments of MDSCs and T cells to demonstrate MDSC immunosuppressive capacity, and interrogate the activation status and cytotoxicity of T cells by staining for IFN γ , TNF α ,

and Granzyme B. All of these data are very convincing. This validation of immune cell function is completely absent from earlier figures, where the authors simply quantify these immune populations and their ratio to make statements about whether a tumor is immunosuppressed. For example, the authors show in Figure 1 that SLC25A22 knockout in APC KRAS mouse tumors leads to decreased MDSC abundance and increased T cell abundance, and they conclude that SLC25A22 knockout reverses KRAS-induced immunosuppression. Without any functional characterization of the activation status of these T cells or their tumor killing capacity ex vivo, no conclusions can be drawn about the immunosuppressive nature of these tumors. All that can be said is that the knockdown altered the T cell abundance. It could be possible that the T cells are more abundant but just as exhausted and dysfunctional and no less suppressed than T cells in tumors without SLC25A22 knockout. The authors should ideally do these functional experiments, or at least alter their language used when describing immune cell infiltration in their models to be less definitive.

Response: We have also performed functional analysis of CD8⁺ T cells in the models in **Fig. 1**. The data were presented in **Fig. 5E-5F** that focused on CD8⁺ T cell function.

Additional points

5. In figure 2C where the authors show their gating strategy to quantify MDSCs and T cells, it's not clear from the density plots shown how the gates were drawn and how the cutoffs were determined. For the CD4 vs CD8 pool in particular, the CD4+ positive and negative populations are not obvious, which can make the gate look somewhat arbitrarily drawn. It would be helpful if the authors overlaid the unstained control sample on these plots to show how the gates were drawn.

Response: We have now overlaid unstained control sample on the plots in **Fig. S3**.

6. Given that response to checkpoint immunotherapy in colorectal cancer varies dramatically based on tumor mutation burden, microsatellite instability, and mismatch repair pathway status, it would be important to know this information about each mouse and human cell line used in this study in order to interpret the anti PD1 treatment data.

Response: Thanks for your comment. We have now added information to the methods section on **p.16, line 504-505**.

7. In the discussion section, the authors primarily reiterate their results and the conclusions they drew from them. There is little to no discussion about where this data fits in the fields of cancer metabolism and cancer immunotherapy, either in relation to previous studies in this area or future directions of the fields. They cite and discuss a single paper from the literature as it related to their findings, Oh et al. 2020. Overall, the authors should put their findings into a larger context.

Response: Thanks for your comment. We have modified the Discussion on **p.14, line 440-449** as follows:

Beyond MDSC, others have reported diverse roles of glutamine in regulating antitumor immunity. JHU083, for instance, enhances oxidative phosphorylation and the antitumor activity of CD8⁺ T-cells¹³, and it synergizes with immunotherapy to achieve total tumor ablation²⁹. V-9302, an inhibitor of glutamine uptake and utilization, suppressed PD-L1 expression in tumor cells, thus promoting anti-PD-L1-mediated antitumor activities of CD8⁺ T-cells³⁰. On the contrary, glutaminase inhibitor CB-839 exerts off-target effects by impairing glutamine metabolism in CD8⁺ T-cells and leading to T-cell suppression³¹. The role of SLC25A22-driven glutamine metabolism in the CXCL1-MDSC crosstalk affirms the function of tumor glutamine metabolism in promoting immunosuppression.

Reviewer #4 (Remarks to the Author):

This manuscript presents a substantial set of experimental evidence that the mitochondrial glutamate transporter SLC25A22 regulates immunosuppression via CXCL1 secretion and recruitment of myeloid-derived suppressor (MDSC) cells in models of KRAS-mutant colorectal cancer (CRC). While glutamine/glutamate metabolism has been linked previously to immunosuppression in syngeneic CRC models (Leone et al. Science 2019), the proposed mechanistic significance of MDSC distinguishes the current work. There is some corroboration in human CRC datasets/cell lines. Another apparently novel aspect of the work is the assertion that the regulation of Src activity by Asn links glutamate metabolism to CXCL1 production in CRC cells. The work is significant in that it potentially offers new routes to reverse the immunosuppressive phenotype of KRAS-driven CRC, a subtype which remains difficult to target. The conclusions are for the most part well supported by the work shown. The most confusing aspect for me relates to the key experiments that first identify Asn as the one downstream metabolite that rescues CXCL1/3 expression/secretion in SLC25A22 deficient cells (Panels 6F/G). I cannot understand, if metabolic production of Asn from glutamate is key, why e.g. extracellular aspartate does not also rescue the phenotype. This is particularly so in light of the subsequent panels where it is shown that interference with the enzyme ASNS (which converts aspartate to asparagine) also inhibits CXCL1 expression – so clearly these cell lines possess sufficient capacity to convert aspartate in the first place. This suggests that some additional experiments may be necessary.

- It could be the case that adding extracellular Asn boosts Src activity and CXCL1/3 production regardless of SLC25A22 status (which would also imply that while losing metabolic uptake glu does reduce Asn it does not fully explain why cytokine production is reduced). The experiments in panel 6F should include what happens to the sgControl and both SLC-KO1/2 in both backgrounds (DLD1, CT26) and perhaps also the parental lines in supplementary. Similarly, Panel 6G should also include what happens to the sgControl lines. Also, more cell permeable versions of α -KG, succinate etc (e.g. the methyl esters) should be used to see if these rescues. The concentrations of the added metabolites should be given in legend/methods.*

Response: We have now performed metabolite addition for both DLD1 and CT26 cells in both sgControl and SLC-KO groups (**Fig. 6F, S10A-S10C**). Under these conditions, asparagine restored CXCL1 mRNA in SLC-KO cells, but had no effect on sgControl cells. Other metabolites had no effect on either (**Fig. 6F, S10A and S10B**). Consistently, asparagine increased CXCL1 secretion in both DLD1 and CT26 SLC-KO cells (**Fig. 6G**) but had no effect on respective sgControl cells (**Fig. S10C**). We have used the cell permeable, methylated forms of α -KG, succinate, OAA, and aspartate (2mM, 24h) and detailed has been added to Figure 6 legend. This information has been added to **p.9, line 286-p.10, line 292**.

- With respect to the experiments above using Asn should be repeated with a titration to the extracellular concentration required to achieve the minimum (low μ M) levels in the cell that are required to active Src (0.6 μ M-6 μ M, Figure 7F). If this intracellular Asn level does not rescue CXCL1/3 expression then it is less likely that Src activation is the link between SLC25A22 and cytokine expression.*

Response: Employing LC-MS with authentic standards, we estimated that intracellular asparagine is in the low μ M (~3 μ mol/mg protein) range in control cells. We have now titrated the lowest concentration of extracellular asparagine (25 μ M) required to restore

intracellular asparagine in SLC25A22 knockout cells (**Fig. S11A**). Consistent with your suggestion, low asparagine fully rescued CXCL1 mRNA and secretion in SLC25A22 knockout cells, and the addition of excess asparagine failed to further increase CXCL1 mRNA and secretion (**Fig. S10A**). This might also explain why asparagine only affects CXCL1 in SLC-KO but not in sgControl cells, as its function might be saturated in the latter cells. This information has been added to **p.10, line 294-298**.

• *It should be proven that the addition of aspartate or cell permeable metabolites do not rescue intracellular Asn concentrations to the level of sgControl and SLC-KO1/2 in both lines (Fig 6H) – if they do, but do not rescue cytokine expression then again it seems unlikely that the link to SLC25A22 mediation of CXCL1/3 holds. The conversion or not of ¹³C labelled aspartate should be reported in the sgControl and SLC-KO lines, and also in the siASNS lines to prove functional interference. The knockdown efficiencies of siASNS in the timeframe of the experiments (in Fig 6J/K) should also be shown with western blot.*

Response: We performed LC-MS analysis of intracellular asparagine, and showed that under identical conditions to that in **Fig. 6H** (2mM, 24h), other metabolites could not restore intracellular asparagine to that of sgControl cells (**Fig. S10D**). We reasoned that the lack of effect of aspartate might be due to low uptake. We thus ectopically expressed aspartate transporter SLC1A3, showing that it facilitated rescue of CXCL1 mRNA and secretion by extracellular aspartate (**Fig. 6F-6G**). This information has been added to **p.10, line 292-294**.

We also performed ¹³C-Aspartate labeling (2mM, 96h) in siASNS and SLC-KO cells with SLC1A3 overexpression. As expected, siASNS inhibited the incorporation of ¹³C-aspartate to ¹³C-asparagine (**Fig. 6I**). No difference was found between SLC-KO and sgControl cells (**Fig. S11C**), as the reduced levels of asparagine is the consequence of reduced intracellular aspartate availability (Wong CC, et al. *Gastroenterology*, 2016), but not the activity of ASNS itself. Finally, we performed western blot to verify ASNS knockdown in **Fig. 6I**. This information has been added to **p.10, line 302-304, 308**.

• *I do not understand in Fig 6J/K when using asparaginase, how in the absence of Asn in the DMEM media the treatment can have an effect i.e. how does it penetrate the cell and lower intracellular Asn to effect Src etc? That does not seem plausible, but metabolomics could be used to prove the effect on intracellular Asn.*

Response: We performed LC-MS analysis of intracellular asparagine, and showed that ASNase *in vitro* inhibited intracellular asparagine (**Fig. 6J**), and it might be a consequence of asparagine efflux after asparaginase treatment. This information has been added to **p.10, line 309**.

• *Figures 7D, E & F – D-asparagine should be used as a control for these experiments to show specificity for the biologically relevant enantiomer and this could be a relevant control for some the experiments above.*

Response: We have now performed experiments in **Fig. 7D-7F** with D-asparagine. D-asparagine did not bind to SRC, as determined by BIAcore (SPR) assay (**Fig. S12A**). In line with this, D-Asparagine also failed to promote SRC kinase activity *in vitro* (**Fig. S12B**). This information has been added to **p.11, line 328-329**.

• *The implication of this paper is in effect that SLC25A22 regulates MDSC via ASNS and Src. Surely more established therapeutics such as Asparaginase and Src inhibitors*

would induce the same effects as targeting SLC25A22? Providing experimental evidence for this in vivo would be important to prove the authors' hypothesis and these key experiments would significantly enhance the translatability of their findings to the clinic.

Response: We have now evaluated the combination of SRC inhibitors or Asparaginase with anti-PD1 in allograft models. We treated MC38K allografts with Dasatinib (SRC inhibitor), anti-PD1, or their combination, and demonstrated that the drug combination of Dasatinib plus anti-PD1 synergistically reduced tumor growth (**Fig. S15A**). Likewise, the combination of Asparaginase injection plus anti-PD1 most effectively suppressed growth of MC38K allografts (**Fig. S15B**). Analyses of tumor infiltrating immune cells showed that combination of ASNase plus anti-PD1 reduced MDSC and PMN-MDSC, whilst increasing CD8⁺ T-cells and IFN- γ ⁺ CD8⁺ T-cells (**Fig. S15C**). These findings collectively enhanced the translatability of our findings to the clinic. This information has been added to **p.12, line 384-388**.

In the discussion it is argued that there would be a therapeutic window for targeting SLC25A22 because other immune cells can use extracellular Asn. However, the authors' own data shows that extracellular Asn can rescue immunosuppressive factor secretion in SLC25A22 negative cells, so how would this approach work in vivo? This weakness should be acknowledged in discussion.

Response: We have now added this limitation to discussion section on **p.15, line 474-476**, as follows:

However, as extracellular asparagine could rescue CXCL1 expression in SLC25A22-depleted tumors, this approach might require further modulation of plasma asparagine levels *in vivo*.

REVIEWERS' COMMENTS

Reviewer #1 (Remarks to the Author):

The authors addressed all reviewer's concerns. No further experiments/clarifications are required.

Reviewer #3 (Remarks to the Author):

The authors have now satisfactorily addressed my concerns, the manuscript is in my view suitable for publication.

Reviewer #4 (Remarks to the Author):

The additional data presented are a very reasonable attempt to address all the concerns raised and increase the level of confidence in the authors' claims. There are also positive implications for the translatability of the work.

Response letter:

Response to Reviewer #1:

The authors addressed all reviewer's concerns. No further experiments/clarifications are required.

Response: We thank the reviewer for the positive comments.

Response to Reviewer #3:

The authors have now satisfactorily addressed my concerns, the manuscript is in my view suitable for publication.

Response: We thank the reviewer for the positive comments.

Response to Reviewer #4:

The additional data presented are a very reasonable attempt to address all the concerns raised and increase the level of confidence in the authors' claims. There are also positive implications for the translatability of the work.

Response: We thank the reviewer for the positive comments.